# Does Deep Learning Learn to Abstract? A Systematic Probing Framework

**Shengnan An**[*†], **Zeqi Lin**[‡], **Bei Chen**[‡], **Qiang Fu**[‡], **Nanning Zheng**[†], **Jian-Guang LOU**[‡]

[†] Institute of Artificial Intelligence and Robotics, Xi'an Jiaotong University

[‡] Microsoft Corporation

`{an1006634493@stu, nnzheng@mail}.xjtu.edu.cn`
`{Zeqi.Lin, beichen, qifu, jlou}@microsoft.com`

## Abstract

Abstraction is a desirable capability for deep learning models, which means to induce abstract concepts from concrete instances and flexibly apply them beyond the learning context. At the same time, there is a lack of clear understanding about both the presence and further characteristics of this capability in deep learning models. In this paper, we introduce a systematic probing framework to explore the abstraction capability of deep learning models from a transferability perspective. A set of controlled experiments are conducted based on this framework, providing strong evidence that two probed pre-trained language models (PLMs), T5 and GPT2, have the abstraction capability. We also conduct in-depth analysis, thus shedding further light: (1) the whole training phase exhibits a "memorize-then-abstract" two-stage process; (2) the learned abstract concepts are gathered in a few middle-layer attention heads, rather than evenly distributed throughout the model; (3) the probed abstraction capabilities exhibit robustness against concept mutations, and are more robust to low-level/source-side mutations than high-level/target-side ones; (4) generic pre-training is critical to the emergence of abstraction capability, and PLMs exhibit better abstraction with larger model sizes and data scales.

## 1 Introduction

*Whereas concrete concepts are typically concerned only with things in the world, abstract concepts are about internal events. — Barsalou et al. (1999)*

Abstraction means capturing the general patterns (often referred to as **abstract concepts**) efficiently in a specific learning context and reusing these patterns flexibly beyond the context (Mitchell, 2021; Kumar et al., 2022; Giunchiglia & Walsh, 1992; Hull, 1920). For instance, the abstraction on language means recognizing the underlying syntax and semantics behind concrete sentences. It is thought to be one of the fundamental faculties in human cognition for effectively learning, understanding and robustly generalizing, and has been studied for a long time in cognitive psychology and behavioral sciences (Gentner & Medina, 1998; Barsalou et al., 1999; Shivhare & Kumar, 2016; Konidaris, 2019).

The abstraction capability is also critical for deep learning, but many previous studies suggested that the surprising success of deep learning may come from the memorization of some surface patterns (also called superficial correlations or shortcuts) (Geirhos et al., 2020; Du et al., 2022), such as some special tokens (Niven & Kao, 2020; Gururangan et al., 2018), overlapping contexts (Lai et al., 2021; Sen & Saffari, 2020), and familiar vocabularies (Aji et al., 2020). It is still unclear whether the models just memorize these patterns without abstractions, or they do learn abstract concepts (yet overwhelmed by surface patterns when applied in a similar context as in training). Therefore, this paper aims to take a step forward to **probe the abstraction capability of deep learning models**, keeping the effects of abstract concepts and surface patterns decoupled and controlled individually.

Our key idea is to probe the abstraction capability from a **transferability** perspective, since surface patterns are always bounded with **task-specific characteristics** while abstract concepts can be more

---

[*]Work done during an internship at Microsoft Research.

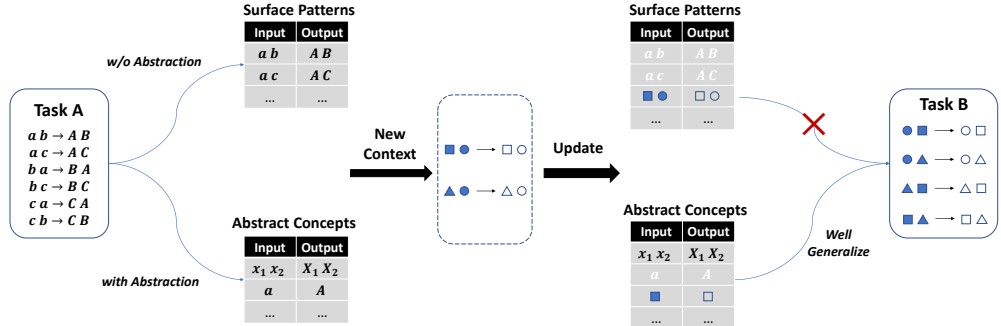

Figure 1: Motivating example: the abstract concepts learned in task $A$ can be effectively reused in task $B$, but surface patterns are useless. Unused patterns or concepts are whitened after the update.

generally reused. We consider designing multiple tasks with shared abstract concepts and totally different surface patterns, then tracing whether the learning on one task can boost the performance on another. Figure 1 demonstrates a motivating example.

**Motivating Example** As shown in Figure 1, suppose we want to examine whether a model can **learn the abstract rule** (i.e., the symbolic mapping rule $x_1 x_2 \rightarrow X_1 X_2$, in which $x_i$ and $X_i$ are general variable slots) from the task $A$, or just **memorize surface maps** (e.g., $ab \rightarrow AB$, in which $a$ and $A$ are task-specific symbols). To reveal the different transferability of two learning mechanisms, we utilize a **probing task** $B$ that contains the same underlying abstract rule as task $A$ but does not overlap with its symbol set. If the model could learn the abstract rule from task $A$, it would reuse it to interpret new context, thus effectively solving task $B$. But if not, memorizing some surface maps that are bounded with task-specific symbols is less effective to solve task $B$.

Motivated by this example, we **design a systematic framework for probing abstraction capability**. This framework considers a set of probing tasks along with three procedures of experiments based on the transfer learning paradigm. The use of abstract concepts and task-specific characteristics in probing tasks are separately controlled. To probe the abstraction capability of language models, this work mainly considers **grammar** as the abstract concept[1]. The grammar of a formal language is a set of hidden rules behind concrete sentences and determines how terminals are combined into sentences that are valid to the syntax. We want to explore **whether the model can be aware of the grammar, or simply memorize some specific word combinations**. We instantiate our framework as a grammar probe that is constructed from the designed formal grammar and terminal sets. The probing results **show strong evidence that two probed PLMs** (specifically, T5-Base (Raffel et al., 2020) and GPT2-Medium (Radford et al., 2019)) **have the abstraction capability** to learn abstract concepts from concrete instances, rather than just simply memorizing surface patterns.

After probing the existence of abstraction capability, we further explore the following questions. **RQ1**: What is the characteristic of the training dynamics on learning abstract concepts? **RQ2**: How are these learned abstract concepts distributed in the model? Concentrated in a few modules or evenly distributed in whole model? **RQ3**: How robust is the abstraction capability on tasks with mutated abstract concepts? **RQ4**: How would generic pre-training and general factors influence abstraction? Here we outline some **interesting findings from our in-depth investigations**: (1) the training phase exhibits a "memorize-then-abstract" two-stage process; (2) the abstract concepts learned in our probes are gathered in a few middle-layer heads; (3) abstraction capability is more robust to source-side/low-level mutations than to target-side/high-level ones; (4) generic pre-training is critical to the emergence of abstraction capability, and larger model size and data scale are beneficial.

**Contributions** 1) We propose a systematic probing framework for abstraction capability, guiding the design of controlled tasks and procedures from a transferability perspective. 2) We instantiate this framework with concrete tasks and show strong evidence that two probed PLMs have the abstraction capability. 3) We further analyze this capability and provide insightful conclusions by investigating the above research questions. Our code and data are publicly available at https://github.com/microsoft/ContextualSP/tree/master/abstraction_probing.

---

[1]We also probed other abstract concepts such as operation semantics in Appendix D.

## 2 RELATED WORK

**Probing deep learning models.** To explore whether deep learning models have certain capabilities, there has been much work examining these black-box models in some specially designed settings, called probes (Petroni et al., 2019; Tenney et al., 2018; Warstadt et al., 2019; Lin et al., 2019; Hewitt & Manning, 2019; Vulić et al., 2020). The key challenge in designing probes is to exclude superficial correlations. That is, the performance of the model in the probing setting should be highly correlated with the capability to be probed rather than other influencing factors. For instance, to probe whether the model encodes some knowledge/information in the representation rather than just over-fit the data, a standard approach is to freeze the model parameters (Petroni et al., 2019; Tenney et al., 2018); to probe whether the model have compositionality rather than just memorize the label distribution, previous work injected statistical bias into the data splits (Lake & Baroni, 2018; Keysers et al., 2019; Kim & Linzen, 2020). In this work, to explore whether models have abstraction capability rather than just memorize surface patterns, we leverage the transferability of abstract concepts, which has been considered as one essential aspect of abstraction (Mitchell, 2021; Kumar et al., 2022) and explored from a cognitive science perspective on neural networks (Dienes et al., 1999; Geiger et al., 2022).

**Abstraction capability.** Abstraction has been studied for a long term in cognitive psychology and behavioral sciences (Hull, 1920; Gentner & Medina, 1998; Barsalou et al., 1999; Burgoon et al., 2013; Wang, 2015; Shivhare & Kumar, 2016; Lake et al., 2017; Daniel, 2017; Konidaris, 2019) and has attracted attention in the artificial intelligence field (Giunchiglia & Walsh, 1992; Richardson et al., 2020; Clark et al., 2020; Talmor et al., 2020; Mitchell, 2021; Zadrozny, 2021; Millhouse et al., 2021; Kumar et al., 2022). The abstraction capability of DNN models has been explored in many tasks such as visual reasoning (Johnson et al., 2017; Barrett et al., 2018; Chollet, 2019; Kumar et al., 2022), grounded language understanding (Ruis et al., 2020), and game playing (Tsividis et al., 2021). As our work focuses on language models, another closely related topic is compositional generalization (Lake & Baroni, 2018; Keysers et al., 2019; Kim & Linzen, 2020), which explored whether neural models could learn high-level grammars from specially designed training examples and apply the learned grammars through compositions. These works concluded that general-propose neural models (such as LSTM and Transformer) could not learn the full grammar with biased observations and demonstrated the importance of symbolic mechanisms for abstraction (Liu et al., 2020; Chen et al., 2020; Liu et al., 2021a). Some other previous work also explored the abstraction of language models in their specially designed tasks (Chollet, 2019; Mitchell, 2021; Zadrozny, 2021).

Most previous explorations of DNN abstraction capabilities did not consider to explicitly avoid and check the influence from task-specific characteristics, thus leaving potential risks that the model may perform well in terms of surface patterns over-fitted to task-specific designs (e.g., patterns in candidate answers (Zhang et al., 2019)) rather than abstract concepts. Some implicit strategies have been leveraged to alleviate such potential influence through indirect ways: some previous work considered using biased task-specific designs in training and test data separately (Kim & Linzen, 2020; Barrett et al., 2018); some have attempted to fix the observed problems in existing probes on an ad hoc basis (Hu et al., 2021; Benny et al., 2021); some considered to inject great task diversity, which implicitly increases difficulty of learning practical shortcut (Chollet, 2019). In this work, rather than implicitly alleviating this potential risks, we consider to explicitly check whether there is performance leakage from surface patterns by leveraging the transferability of abstraction capability and comparing performance among a set of controlled experiments.

## 3 PROBING FRAMEWORK

As mentioned in Section 1, **abstraction** is the capability to induce **abstract concepts** from concrete instances in a certain learning context and flexibly generalize these concepts beyond the context. A key difference between a surface pattern and an abstract concept is their different cross-task transferability, as the former is always bounded with some task-specific characteristics (e.g., a certain vocabulary) while the latter is transferable across tasks. We define this property as following.

**Property: Transferability of Abstract Concepts.** Consider two machine learning tasks $A$ and $B$ that do not share any common instances between their task-specific characteristics spaces, but have essentially the same set of abstract concepts behind them, the transferability of abstract concepts means that learning $A$ can help better learn $B$.

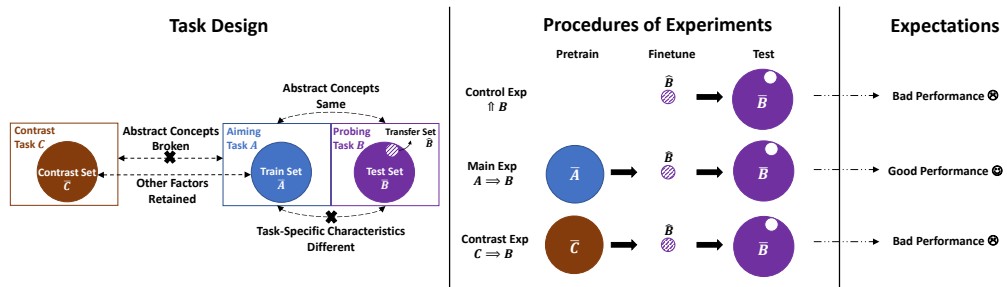

Figure 2: The illustration of the probing framework.

**Notations:** We denote two learning paradigms: $A \Rightarrow B$ as a procedure that the model is first trained on $A$ and then fine-tuned on $B$, and $\Uparrow B$ as a procedure of directly training on $B$ without the pre-training on $A$. The transferability can be evaluated with the performance gain from procedure $A \Rightarrow B$ compared to $\Uparrow B$, denoted as $\Delta(A \Rightarrow B)$.

Based on this property, **we can verify the learning of abstract concepts by checking whether the transferability is exhibited** (i.e., assessing $\Delta(A \Rightarrow B)$). In the following, we design a framework for probing the learning of abstract concepts in a systematic manner and illustrate it in Figure 2.

**Aiming**

- This framework examines whether a probed model could learn abstract concepts $\mathbb{C}_A$ from the **aiming task** $A$ with a **train set** $\overline{A}$.

**Task Design**

- **Probing task** $B$ with the **transfer set** $\hat{B}$ and **test set** $\overline{B}$ contains the abstract concepts $\mathbb{C}_B$ that is required to be the same as $\mathbb{C}_A$. The task-specific characteristics used to construct $\hat{B} \cup \overline{B}$ do not overlap with that of $\overline{A}$. In addition, the examples in $\hat{B}$ are restricted to contain insufficient information for the probed model to learn $\mathbb{C}_B$ perfectly. Thus, the gain from the abstraction in task $A$ would be noticeable.
- **Contrast task** $C$ with the **contrast set** $\overline{C}$ aims to further confirm that the performance in task $B$ is principally correlated with abstractions rather than other factors. The abstract concepts $\mathbb{C}_C$ is constructed by greatly breaking (changing) $\mathbb{C}_A$, thus compared with task $A$, the abstraction on task $C$ is less effective for solving task $B$. The task-specific characteristics and other latent factors in constructing $\overline{C}$ are kept the same with that in $\overline{A}$.

**Procedures of Experiments** (abbreviated as Exp)

- **Control Exp** $\Uparrow B$**:** only train the model on $\hat{B}$ and test on $\overline{B}$.
- **Main Exp** $A \Rightarrow B$**:** train the model on $\overline{A}$, then fine-tune on $\hat{B}$, finally test on $\overline{B}$.
- **Contrast Exp** $C \Rightarrow B$**:** train the model on $\overline{C}$, then fine-tune on $\hat{B}$, finally test on $\overline{B}$.

**Hypothesis and Expectations**

- **Hypothesis:** the probed model can learn abstract concepts $\mathbb{C}_A$ from $\overline{A}$.
- **Expectation 1:** $\Delta(A \Rightarrow B)$ is significantly high, i.e., $A \Rightarrow B$ brings considerable gain compared with $\Uparrow B$.
- **Expectation 2:** $\Delta(C \Rightarrow B)$ is significantly lower than $\Delta(A \Rightarrow B)$ (or close to zero), i.e., Expectation 1 is highly correlated with the learning of abstract concepts rather than other factors.

## 4 PROBING ABSTRACTION CAPABILITY OF LANGUAGE MODELS

The abstract concepts mainly considered in this work is grammar, a set of syntactic rules hidden behind concrete sentences that determine how terminals are combined into sentences that are valid to

Table 1: Part of $\mathbb{G}_s$ and $\mathbb{G}_t$. Rules in the last row are allowed to iterate up to 12 times.

| Source-Side Grammar $\mathbb{G}_s$ | Target-Side Grammar $\mathbb{G}_t$ | Type |
|---|---|---|
| verb $\twoheadrightarrow$ $\mathbf{S}_v$
sub / direct-obj / indirect-obj $\twoheadrightarrow$ $\mathbf{S}_n$
conj $\twoheadrightarrow$ $\mathbf{S}_c$ | PREDICATE $\twoheadrightarrow$ $\mathbf{S}_P$
AGENT / THEME / RECIPIENT $\twoheadrightarrow$ $\mathbf{S}_E$
CONCAT $\twoheadrightarrow$ $\mathbf{S}_C$ | T-Production Rule |
| sentence $\twoheadrightarrow$ subj verb
sentence $\twoheadrightarrow$ subj verb direct-obj indirect-obj
sentence $\twoheadrightarrow$ sentence conj sentence | CLAUSE $\twoheadrightarrow$ PREDICATE ( AGENT )
CLAUSE $\twoheadrightarrow$ PREDICATE ( AGENT, THEME, RECIPIENT )
CLAUSE $\twoheadrightarrow$ CLAUSE CONCAT CLAUSE | N-Production Rule |

the syntax. To design a grammar probe, we instantiate the framework with formal language translation (FLT) tasks. We assume that **the generative grammar of the source and target languages contain the abstract concepts of FLT tasks**, and that the surface patterns (e.g., familiar bigrams) are bounded with task-specific terminal sets. We give a more specific definition of abstraction based on FLT tasks:

**Definition**: Considering an FLT task $T : \mathcal{L}_s \rightarrow \mathcal{L}_t$ that translate the source language $\mathcal{L}_s$ (with grammar $\mathbb{G}_s$ and terminals $\mathbf{S}_s$) to the target language $\mathcal{L}_t$ (with grammar $\mathbb{G}_t$ and terminals $\mathbf{S}_t$), and a set of concrete pairs $\overline{T} = \{(l_s^i \rightarrow l_t^i)\}^k$ in which $l_s^i$ and $l_t^i$ are sentences from $\mathcal{L}_s$ and $\mathcal{L}_t$ respectively, the abstraction capability is learning the map from $\mathbb{G}_s$ to $\mathbb{G}_t$ during training on $\overline{T}$ rather than just simply memorizing terminal-specific patterns that are bounded with $\mathbf{S}_s$ and $\mathbf{S}_t$.

Our FLT tasks are majorly derived from the synthetic semantic parsing task COGS (Kim & Linzen, 2020) and the Probabilistic Context-Free Grammar (PCFG) it used. We directly take the source grammar $\mathbb{G}_s$ in COGS which mimics the English natural language grammar, and reconstruct the target grammar $\mathbb{G}_t$ in COGS to be chain-structured (detailed in Appendix K.1). The map from $\mathbb{G}_s$ to $\mathbb{G}_t$ is a homomorphism (partly shown in Table 1). Terminals can be divided into three groups: the verbs $\mathbf{S}_v$ in $\mathbb{G}_s$ (aand the PREDICATEs $\mathbf{S}_P$ in $\mathbb{G}_t$), the nouns $\mathbf{S}_n$ (the ENTITYs $\mathbf{S}_E$) and the conjunctions $\mathbf{S}_c$ (the CONCATs $\mathbf{S}_C$). The production rules can be categorized as T-Production rules (only containing terminals at the right side) and N-Production rules.

We assign to the tasks $A$ and $B$ the same set of production rules while different terminals. It means that task $A$ and $B$ share the same abstract concepts while having no overlap between the task-specific characteristic spaces. For constructing task $C$, we completely change the production rules for $A$ while preserving the terminal sets, thus task $A$ and $C$ do not share abstract concepts while could have similar task-specific characteristics. We describe the instantiation of different sets in detail as follows. Examples in these sets are contained in Appendix F.1.

**Train set $\overline{A}$.** To generate examples in $\overline{A}$, we derive $\mathbb{G}_s^+$ and $\mathbb{G}_t^+$ by only one-to-one replacing the terminals in $\mathbb{G}_s$ and $\mathbb{G}_t$ with new ones[2]. New terminals are sampled from the Wordlist Corpora in NLTK (Bird et al., 2009). Additionally, as the original $\mathbf{S}_c$ (also $\mathbf{S}_C$) only contains a single terminal, we add 31 additional terminals into the new $\mathbf{S}_c$ (and $\mathbf{S}_C$) to increase the diversity. The terminal diversity will be further discussed in Section G.1.

**Transfer set $\hat{B}$ and Test set $\overline{B}$.** We take the train set in COGS as $\hat{B}$, and take the sentential complement (*Com.*) set and subject modification (*Mod.*) set as $\overline{B}$ for two sub-probes. The $\hat{B}$ only contains examples with up to 2 recursions and object modifications, while the $\overline{B}$ contains up to 12 recursions and subject modifications. It has been proved that training on $\hat{B}$ is not enough for a DNN model to learn the full grammars of COGS for handling the test cases in $\overline{B}$ (Kim & Linzen, 2020).

**Contrast set $\overline{C}$.** Compared with $\overline{A}$, $\overline{C}$ is generated with the same source grammar $\mathbb{G}_s^+$, but the target grammar is totally changed as $\mathbb{G}_t^-$: for each rule of $\mathbb{G}_t^+$, its right-side word order is reversed[3]. Except for the generative grammar, all other factors are kept the same with $\overline{A}$ during generating $\overline{C}$.

## 5 Experimental Setup and Main Results

We probe two pre-trained language models: T5-Base and GPT2-Medium. Our experiments are based on the Huggingface Transformer models (Wolf et al., 2020). For both (continue) pre-training and fine-tuning, we take Adam (Loshchilov & Hutter, 2018) with 1e-5 learning rate and 0.01 weight

---

[2]Some non-semantic terminals are kept the same, such as the period in $L_{src}$ and parentheses in $L_{tgt}$.

[3]Some basic rules are preserved (e.g., the order of the preceding and following parentheses).

Table 2: The main results of our probe. $\Delta(A \Rightarrow B)$ and $\Delta(C \Rightarrow B)$ are in brackets. The evaluation metric is exact match accuracy (%). *Com.* and *Mod.* represent the sentential complement and subject modification sets for $\overline{B}$. These results are in line with our two expectations.

| Model | Sub-Probe | Control Exp $\Uparrow B$ | Main Exp $A \Rightarrow B$ | Contrast Exp $C \Rightarrow B$ | Model | Sub-Probe | Control Exp $\Uparrow B$ | Main Exp $A \Rightarrow B$ | Contrast Exp $C \Rightarrow B$ |
|---|---|---|---|---|---|---|---|---|---|
| | Avg. | 18.7 | 71.9 (+53.2) | 16.0 (-2.7) | | Avg. | 8.0 | 47.9 (+39.8) | 8.1 (+0.1) |
| T5 | *Com.* | 23.1 | 88.2 | 15.4 | GPT2 | *Com.* | 1.9 | 48.2 | 2.6 |
| | *Mod.* | 14.3 | 55.6 | 16.5 | | *Mod.* | 14.1 | 47.6 | 13.6 |

decay. Batch size is 8 and max training step is 100k. We generate 3 groups of new terminals, repeat the experiments on each group with 2 random seeds, and finally average 6 results. The early-stopping strategy is applied to avoid catastrophic forgetting. Detailed settings are listed in Appendix K.

Table 2 shows the main results of our probe. For both two sub-probes, the performances of two probed models are **in line with two Expectations** set in Section 3. First, the results of $\Uparrow B$ are very low, and $A \Rightarrow B$ can bring significant improvement, which is in line with Expectation 1. Second, the results of $C \Rightarrow B$ are much lower than $A \Rightarrow B$ (and are even just comparable with $\Uparrow B$), which is in line with Expectation 2. As two expectations are experimentally examined, we can draw a preliminary conclusion: **our probing results provide strong evidence that two probed PLMs have the abstraction capability** to learn abstract concepts from concrete instances rather than just memorize surface patterns, and to transfer the learned abstract concepts beyond specific tasks.

## 6 ANALYSIS

Based on our designed probe and results above, we further analyze the abstraction capability of PLMs to answer the RQs mentioned in Section 1. All experiments below are derived from *Com.* sub-probe, and are mainly conducted with T5-Base model except that are explicitly mentioned.

### 6.1 LEARNING PROCESS OF ABSTRACT CONCEPTS

To investigate the **learning process of abstract concepts**, we save checkpoints for every 1,000 steps during training on $\overline{A}$. Each checkpoint is further fine-tuned on $\hat{B}$ and tested on $\overline{B}$. For comparison, we also investigate **the process of memorizing surface patterns** by directly examining each checkpoint on the held-out dev set in task $A$. Figure 3 shows the performance curves of two learning processes.

**The training phase exhibits a "memorize-then-abstract" two-stage process.** As shown in Figure 3, there is an obvious phase difference (48k training steps) between two time points that two learning processes achieve their 90% relative performance, respectively. Such a phase difference means that when the model has already performed well on task $A$ in an early training phase, the learning of desired abstract concepts is still on-going. In other words, the in-task performance in an early training phase comes mainly from the **effects of some task-specific surface patterns** rather than general abstract concepts. With extending the training phase, the abstract concepts can be further extracted/enhanced. This phase difference also suggests that **the pre-training process should be continued even if the model has already achieved a good in-pre-training performance**.

**The learning of abstract concepts is accelerated after in-task examples are well learned.** After the model reaches 90% in-task relative performance (i.e. right side of the red dashed line), the learning curve of abstract concepts (i.e., the blue curve) rises more rapidly.

**The learning of abstract concepts is not stable in the early training phase.** The curve of in-task performance is much smoother than the cross-task one. This suggests that the learning and transfer of abstract concepts is not stable. Nevertheless, the large fluctuations occur mainly in the early phases of training. With increasing training steps, this instability gradually decreases.

### 6.2 ABSTRACT ATTENTION HEADS

To investigate how the learned abstract concepts are distributed in the model, we first conduct preliminary experiments by separately freezing parameters in each layer and sub-layer during fine-tuning, and find that the parameters in attention sub-layers play important roles (detailed in Appendix E). To

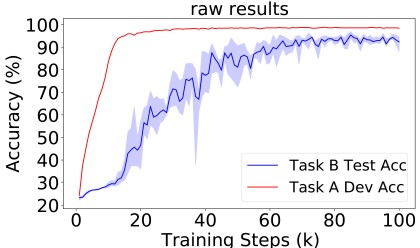 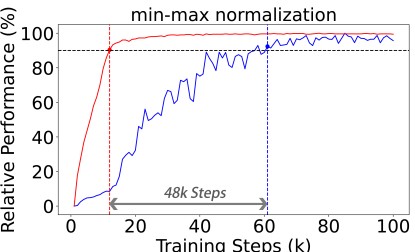

Figure 3: Two learning process. Blue curves represent the learning performance of abstract concepts and red curves represent the learning performance of in-task examples.

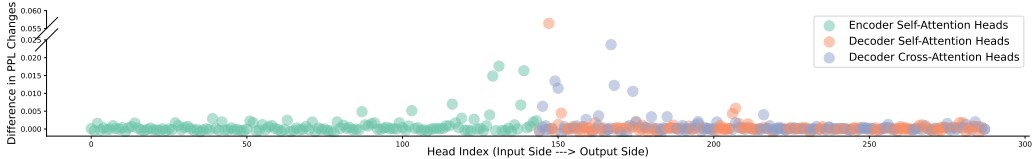

Figure 4: DPC of each pruned head. The heads sorted from left to right are located from the first to the last layer in the model.

further determine the contribution of each attention head, we consider measuring the performance degradation after excluding the effect of each head. Specifically, we evaluate the change in perplexity (PPL) of examples in $\overline{B}$ after pruning the normalized wight of each head as follows,

$$\Delta_{\theta,\overline{B}}(h) = \frac{1}{|\overline{B}|} \sum_i [\text{PPL}(l_t^i|l_s^i; \theta_{-h}) - \text{PPL}(l_t^i|l_s^i; \theta)], \tag{1}$$

in which $h$ represents a certain head, $\theta$ is the full set of parameters in PLM after fine-tuning on $\hat{B}$, $\theta_{-h}$ means pruning the $h$ head, and $(l_s^i, l_t^i)$ is the input-output pair in $\overline{B}$. Note that a higher PPL means a lower performance. Considering that some heads may store the task-specific knowledge learned from fine-tuned data $\hat{B}$, pruning these heads may also lead to performance changes. Therefore, we also evaluate a baseline PPL change $\Delta_{\theta,\hat{B}}(h)$ on fine-tuned examples in $\hat{B}$ and measure the difference in PPL changes (DPC)$= \Delta_{\theta,\overline{B}} - \Delta_{\theta,\hat{B}}$. The DPC of each head is shown in Figure 4.

**Abstract concepts are largely contained in a few heads, not evenly distributed in all heads.** Note that there are totally 432 attention heads in T5-Base. Figure 4 shows that among hundreds of heads, only a dozen of them are highly correlated with storing abstract concepts in our probe.

**These abstract attention heads are gathered in middle layers in T5.** A larger index in Figure 4 means that the corresponding head is more away from the input side and closer to the output side. It shows that the abstract attention heads (i.e., heads with high DPC) are mainly located in the middle layers of T5-Base model, i.e., the last encoder layers and first decoder layers.

We further explore whether abstract concepts are modularized in the model. A module is a part of parameters that can individually perform a specific target functionality (Csordás et al., 2020). To investigate modularity, we take the method of freezing certain parameters during fine-tuning to examine whether the update of these parameters can be independent. We consider the top 36 heads with the highest DPC (which contain some redundant heads) as abstract heads. For comparison, we separately experiment with freezing 36 random heads. Table 3 shows that freezing abstract heads takes effect while freezing random heads does not. We further explore the modularity in Appendix E.

Table 3: Freeze abstract heads.    Table 4: PLMs performance with fuzzy abstract concepts.

| Method | $A \Rightarrow B$ |
|---|---|
| Baseline | 92.8 |
| +Freeze Abstract Heads | 96.6 (+3.8) |
| +Freeze Random Heads | 92.9 |

| Probe | Model | Control Exp $\Uparrow B$ | Main Exp $A \Rightarrow B$ | Contrast Exp $C \Rightarrow B$ |
|---|---|---|---|---|
| Fuzzy Grammar | T5 | 24.0 | 35.1 (+11.1) | 26.2 (+2.2) |
| | GPT2 | 16.4 | 21.0 (+4.6) | 11.6 (-4.8) |

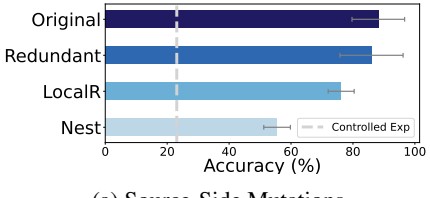
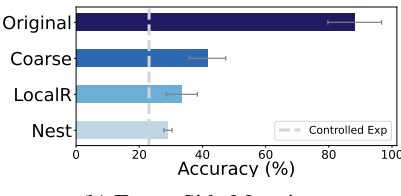

(a) Source-Side Mutations          (b) Target-Side Mutations

Figure 5: Performance with different derivations of (a) source and (b) target grammar.

## 6.3 ROBUSTNESS OF ABSTRACTION CAPABILITY

We explore the robustness of the probed abstraction capability when the abstract concepts in our designed probes are mutated. Different from the contrast task $C$ in which the target grammar is totally changed, here we consider partially injecting mutations into source/target-side grammar. According to the formal grammar in Table 1, we consider injecting mutations at different abstract levels: changing T-Production rules can be regarded as a **low-level mutation**, since only terminals will be influenced and the whole sentence structure is kept; changing non-iterative N-Production rules can be regarded as a **mid-level mutation**, since the local structure will be mutated but the whole recursive structure is preserved; changing iterative N-Production rules can be regarded as a **high-level mutation**, since the whole recursive structure will be reconstructed. Based on the grammar used in formal language task $A$, we design three derivations $\mathbb{G}_t^*$ by injecting mutations into the original $\mathbb{G}_t^+$ in different levels. Examples of different derivations are contained in Appendix F.4.

**Coarse $\mathbb{G}_t^*$ (low-level mutation):** We omit the non-terminals AGENT, THEME, RECIPIENT, and their corresponding terminals. In other words, the second T-Production rule in Table 1 is removed. Compared with $\mathbb{G}_t^+$ (also $\mathbb{G}_t$), Coarse $\mathbb{G}_t^*$ does not contain detailed arguments of PREDICATE.

**Local Reverse (LocalR) $\mathbb{G}_t^*$ (mid-level mutation):** The local word order in a sentence is reversed. Specifically, we reverse the right-side word orders of the N-Production rules, except for the rule in the last row of Table 1 which is an iterative one. It means that the order of CLAUSEs (determined by the last rule) remains the same, while the terminals in each CLAUSE are locally reversed.

**Nested $\mathbb{G}_t^*$ (high-level mutation):** It is obtained by changing the iterative rule (i.e, the last rule in Table 1) from the chain-structure to be nested. The new N-Production rule is "CLAUSE → PREDICATE ( AGENT, CONCAT CLAUSE )".

We can also construct $\mathbb{G}_s^*$ from $\mathbb{G}_s^+$ with the same technique except for the coarse one, as the source language must contain enough information to generate targets. Thus, we design a Redundant $\mathbb{G}_s^*$ which contains redundant terminals that are not mapped into targets (detailed in Appendix F.3). We separately change the source and target grammars to derivations and show results in Figure 5.

**PLMs can exhibit robustness against mutations in abstract concepts.** Results of these derivations with mutations are higher than the Control Exp (and Contrast Exp), indicating that even though the learned abstract concepts are only partially matched with that in downstream tasks, the abstraction capability of PLMs can still leverage the similar parts in two sets of mutated abstract concepts.

**Abstraction capability is more robust to low-level mutations.** Among three kinds of derivations, the low-level mutated ones (Coarse $\mathbb{G}_t^*$ and Redundant $\mathbb{G}_s^*$) perform best, while the high-level mutated ones (Nested $\mathbb{G}_t^*$ and $\mathbb{G}_s^*$) perform worst. This trend implies that the robustness of the abstraction capability decreases as the mutation level of abstract concept rises. This also suggests that matching of high-level abstract concepts should be prioritized when selecting pre-training tasks.

**Abstraction capability is more robust to source-side mutations.** Comparing the results in Figure 5a and 5b, source-side mutations bring less affects to downstream performance than target-side ones, indicating that PLMs can more robustly reuse source-side abstract concepts.

**Redundant information barely affects abstraction.** Surprisingly, the performance of Redundant $\mathbb{G}_s^*$ is nearly the same with that of the original $\mathbb{G}_s^+$, which means that injecting redundant information into inputs would hardly affect the learning of abstract concepts. It indicates that the abstract capability of PLM can naturally exclude the influence of irrelevant information.

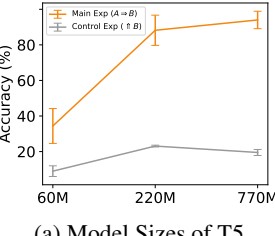 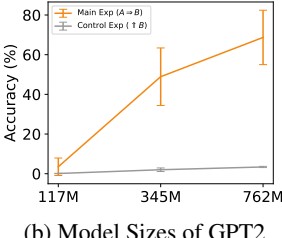 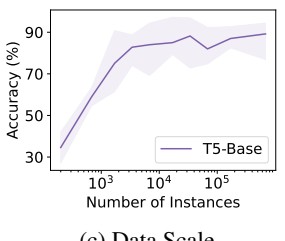

(a) Model Sizes of T5      (b) Model Sizes of GPT2      (c) Data Scale

Figure 6: Performance with different model sizes, data scales and data diversity.

Table 5: Probing results for randomly initialized models. $\Delta(A \Rightarrow B)$ and $\Delta(C \Rightarrow B)$ are in brackets.

| Model | Sub-Probe | Control Exp $\Uparrow B$ | Main Exp $A \Rightarrow B$ | Contrast Exp $C \Rightarrow B$ | Model | Sub-Probe | Control Exp $\Uparrow B$ | Main Exp $A \Rightarrow B$ | Contrast Exp $C \Rightarrow B$ |
|---|---|---|---|---|---|---|---|---|---|
| | Avg. | 4.7 | 6.7 (+2.0) | 5.8 (+1.1) | | Avg. | 5.1 | 6.8 (+1.7) | 4.7 (-0.4) |
| T5 | *Com.* | 0.1 | 1.0 | 0.1 | GPT2 | *Com.* | 0.1 | 1.5 | 0.4 |
| | *Mod.* | 9.3 | 12.3 | 11.6 | | *Mod.* | 10.1 | 12.0 | 9.0 |

**Fuzzy abstract concepts can also be learned and transferred.** Compared with the formal grammar discussed above, which can be concretely defined, fuzzy grammar is more free (such as natural language grammar). To explore how would abstraction capability perform on fuzzy grammar, we take natural language sentences for experiments and design different sets by mimicking the *Com.* sub-probe. Detailed designs are described in Appendix H. We report BLEU score in Table 4. It shows that the performance of PLMs on learning fuzzy grammar is also in line with our two expectations.

## 6.4 GENERAL FACTORS & GENERIC PRE-TRAINING

As explored in previous work, there are some general factors that influence the performance of DNN models (Bommasani et al., 2021; Wei et al., 2022; Henighan et al., 2020), such as model size and data scale. We investigate how these general factors and the generic pre-training influence the learning of abstract concepts. More results and analysis can be found in Appendix G.1.

**PLMs exhibit better abstraction with larger model sizes.** Figure 6a and 6b show that for both T5 and GPT2 architectures, larger pre-trained language models have better abstraction capability than the smaller ones, as we can observe that the gains from the Control Exp to the Main Exp become greater with the model sizes increasing.

**Larger data scale in pre-training helps better exhibit abstraction.** Figure 6c shows T5-Base performance with different scales of the train set $\overline{A}$. It shows that performance increases rapidly from ~300 to ~3.4K (with ~50% absolute accuracy improvement) and improves marginally (and unstably) from ~3.4K to ~680K (with ~5% absolute accuracy improvement). Overall, the performance trend is going up with data scale increasing, indicating that the larger data scales benefit abstraction.

**Generic pre-training is critical for the emergence of abstraction.** We probe the abstraction capability of randomly initialized T5-Base and GPT2-Medium (i.e., without loading pre-trained checkpoints) and report the results in Table 5. The poor performance on $A \Rightarrow B$ reveals that without generic pre-training, these deep learning models can hardly extract transferable abstract concepts from task $A$, even though they can still achieve >98% dev set performance on task $A$ by fitting some task-specific suffer patterns. The comparison of the results in Table 2 and Table 5 demonstrate that the broader background pre-training is critical for the emergence of abstraction capability.

## 7 CONCLUSIONS

In this paper, we introduce a systematic probing framework from a transferability perspective to guide the design of probes for abstraction capability. We instantiate this framework as a grammar probe and show strong evidence that two probed PLMs have the abstraction capability. We further analyze this probed capability by investigating several in-depth questions and provide insightful conclusions.

ETHICS STATEMENT

A sufficiently robust abstraction capability that can perfectly extract abstract concepts and exclude concrete information in any situation will help deep learning models avoid many potential risks of ethical issues such as social bias and privacy breaches. However, as investigated in this work, the abstraction capability of some commonly used deep learning models may be fragile and can be affected by their training situation. This suggests that the abstraction capabilities of these models are still not reliable enough to naturally avoid these potential ethical issues, and calls for future work to explore ways to strengthen the robustness of the abstraction capabilities of deep learning models.

ACKNOWLEDGMENTS

We thank all the anonymous reviewers for their valuable comments. This work was supported in part by NSFC under grant No. 62088102. We would like to thank Qian Liu for his valuable suggestions and feedback on this work.

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

This is the Appendix of the paper *Does Deep Learning Learn to Abstract? A Systematic Probing Framework*.

## A  DISCUSSIONS

Below are more discussions about our work.

**Potential factors that may hinder our probing.**   We consider the main factor that could block the use of our probing framework is the catastrophic forgetting problem in deep learning (Goodfellow et al., 2013; Kemker et al., 2018). Since our probing framework relies on the transferability property of abstract concepts, if catastrophic forgetting dominates the learning of downstream tasks, such transferability will hardly take effect and the probing results will fail to reveal the abstraction capabilities. Considering this problem, we utilize the early-stopping strategy (detailed in Appendix) to alleviate catastrophic forgetting. Moreover, our tested pre-trained models are naturally more robust to catastrophic forgetting (Ramasesh et al., 2021).

**Better understanding "why does transfer learning work".**   Recent success of transfer learning shows that pre-training (or continue pre-training) with similar source tasks can help better solve downstream target task (e.g., question answering (Khashabi et al., 2020; Liu et al., 2021b), face verification (Cao et al., 2013), and general NLU tasks (Pruksachatkun et al., 2020)). Some previous work in cross-lingual transfer learning empirically observed that the model can transfer some knowledge beyond vocabulary (Artetxe et al., 2020; Ri & Tsuruoka, 2022), but they did not consider to exclude the influence from other potential factors. Our results can serve as stronger evidence for the reason to the success of transfer learning, that in addition to transferring some surface patterns, the better target performance can also benefit from similar abstract concepts learned from source tasks.

**Limitations and future work.**   The main limitations in this work are 1) we do not quantify the abstraction capability and 2) we only test two large pre-trained models. We leave these two points to our future work. Another future direction is to further explore the mechanisms behind abstractions.

## B  COMPARISONS WITH PREVIOUS FINDINGS ABOUT LEARNING DYNAMIC

**Comparison with information bottleneck.**   Shwartz-Ziv & Tishby (2017) found a two-phase learning process from the view of information flow in deep neural networks: empirical error minimization phase and representation compression phase. This process is different from the memorize-then-abstract process since they measure the training dynamics in quite different perspectives. The former focuses on the compression of representation (and reduction of mutual information) while the latter portrays the learning of abstract concepts. The analogy between the two may lie in that the extraction of abstract concepts from concrete instances is in some way have the effect of information compression.

**Comparison with Grokking.**   Power et al. (2022) revels that the improvement in generalization (on validation set) can happen well past the point of over-fitting (on train set). Both 'grokking' and 'memorize-then-abstract' phenomenon indicate that some general patterns are always learned in a later training stage. The difference is that the 'grokking' focuses on generalization beyond over-fitting training data, while 'memorize-then-abstract' portrays the transfer of abstract concepts beyond task-specific characteristics.

## C  PRINCIPLES FOR DESIGNING PROBING TASKS

To verify whether the model could learn abstract concepts from task $A$ by assessing $\Delta(A \Rightarrow B)$, we propose the following principles for designing task $B$:

1) The space of task-specific characteristics of $B$ should be very different from that of $A$ so that the memorization of surface patterns in $A$ is helpless to $B$.
2) The abstract concepts of $B$ should be the same as that of $A$ so that the abstraction on $A$ could be reflected with a better performance on $B$.
3) The fine-tuning-only approach $\Uparrow B$ should be not enough to learn task $B$ perfectly; otherwise, the gain from the abstraction on $A$ would not be noticeable.

Furthermore, to verify that the performance gain on $B$ is from the abstraction on $A$ rather than other factors, we consider a contrast task $C$ of $A$:

4) The abstract concepts of $C$ should be very different from $A$ (also $B$), while other latent factors are kept the same such as data scale and pre-training steps.

The consideration of contrast task is similar to *selectivity* (Hewitt & Liang, 2019).

## D   AN OPERATION PROBE

The operations semantics (e.g., conjunction) in the Boolean algebra can be regarded as transition functions between the given Boolean variables and corresponding outputs. We want to examine **whether the model can learn the meaning of operations from concrete logical expressions, or just learn superficial correlations from specific sketches in expressions**.

In operation probe, we instantiate the framework with logical expression evaluation (LEE) tasks. We consider **operation semantics in logical expressions as abstract concepts**, and surface patterns (e.g., local string matching) are bounded with operation sketches.

Figure 7a shows two kind of sketches: chain sketch and tree sketch. The model trained on chain sketch may learned the meaning of operations (e.g., conjunction of two Boolean variables) or simply memorize some head or tail patterns of strings (e.g., if the head of input string is "False AND ( ", the output is always "False"). Learning operation semantics can more generally help understand other expressions with different sketches, but memorizing head or tail patterns in chain sketches is helpless or even harmful to understand tree sketches, since these surface patterns can lead to wrong results in different sketches. We give a more specific definition of abstraction based on LEE tasks:

**Definition 2**: Considering an LEE task $L : \mathcal{E}_s \to \mathcal{B}_t$ that the source logical expressions $\mathcal{E}_s$ (with operation semantics $\mathbb{P}_s$ and sketch $\mathbf{K}_s$) are evaluated as Boolean values $\mathcal{B}_t$, and a set of input-output pairs $\overline{L} = \{(e_s^i \to b_t^i)\}^k$ in which $e_s^i$ is an logical expression sampled from $\mathcal{E}_s$ and $b_t^i \in \{True, False\}$ is the evaluation result of $e_s^i$, the abstraction capability is learning the meanings of operations $\mathbb{P}_s$ during training on $\overline{L}$ rather than memorizing sketch-specific patterns that are bounded with $\mathbf{K}_s$.

Our LLE tasks probe the learning of four logical operations: $\mathbb{P}_s^+ =$ {Conjunction (Conj.), Disconjunction (Disc.), Alternative Denial (Alt.), Joint Denial (Joi.)}. Figure 7b illustrates the transition functions of these operations.. For generating data, each operations in $\mathbb{P}_s^+$ is constantly aligned with one operator (i.e., a concrete string) in $\mathbf{S}_o$. Examples of these sets are contained in Appendix F.2.

**Train set $\overline{A}$.** We synthesize the data in $\overline{A}$ with $\mathbb{P}_s^+$ and chain sketch. Each expression contains eight operators which are sampled from $\mathbf{S}_o$.

**Transfer set $\hat{B}$ and Test set $\overline{B}$.** We synthesize the data in $\hat{B} \cup \overline{B}$ with $\mathbb{P}_s^+$ and tree sketch. Each expression contains eight operators which sampled from $\mathbf{S}_o$. When probing a certain operation $\mathbf{p}_s \in \mathbb{P}_s^+$, we limit that the expression in $\hat{B}$ does not contain $\mathbf{p}_s$, while each expression in $\overline{B}$ must contain $\mathbf{p}_s$. To make the model familiar with operators in $\mathbf{S}_o$ or not forget them during further fine-tuning, we supplement $\hat{B}$ with 100 two-operator expressions which cover the full $\mathbf{S}_o$. Empirically, as DNN models can be easily influenced by the statistical bias in label distributions, we balance the 'True' and 'False' examples during sampling.

**Contrast set $\overline{C}$.** The operators and sketch in $\overline{C}$ are kept the same with $\overline{A}$, but each operator in $\mathbf{S}_o$ is aligned with another set of logical operations $\mathbb{P}_s^- =$ {Material Implication, Converse Implication, Material Non-implication, Converse Non-implication}. The transition functions of these operations are listed in Appendix F.2.

Results of our operation probe is shown in Table 6.

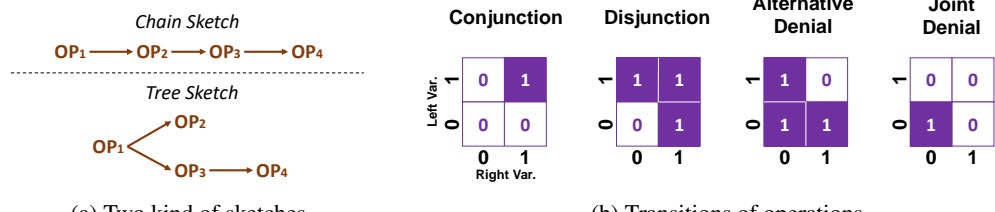

(a) Two kind of sketches.        (b) Transitions of operations.

Figure 7: Four operations and two sketches in our operation probe. (a) shows the chain sketch used in task $A$ and $C$, and the tree sketch used in $\hat{B} \cup \overline{B}$. Each 'OP' represents one operation. (b) shows the transition results of four operations with different left and right Boolean variables.

Table 6: Results in our operation probe.

| Probe | Model | Sub-Probe | Control Exp $\Uparrow B$ | Main Exp $A \Rightarrow B$ | Contrast Exp $C \Rightarrow B$ |
|---|---|---|---|---|---|
| Operation | T5 | Avg. | 70.1 | 88.9 (+18.8) | 71.2 (+1.1) |
| | | Conj. | 63.3 | 90.8 | 65.3 |
| | | Disc. | 58.9 | 84.8 | 72.8 |
| | | Alt. | 79.1 | 90.7 | 72.6 |
| | | Joi. | 79.2 | 89.2 | 74.3 |
| | GPT2 | Avg. | 64.7 | 76.3 (+11.6) | 64.0 (-0.7) |
| | | Conj. | 61.8 | 78.6 | 61.1 |
| | | Disc. | 59.8 | 72.5 | 64.1 |
| | | Alt. | 69.0 | 78.6 | 65.4 |
| | | Joi. | 68.3 | 75.3 | 65.3 |

# E ABSTRACT CONCEPTS ARE MODULARIZED IN PLMS

To supplement our analysis on abstract attention heads, here we provide our detailed explorations to identify the modulars in PLMs that store abstract concepts in our probes. Our explorations are based on the following property and assumption.

**Property: Forgetting of Abstract Concepts.** Consider the Main Exp $A \Rightarrow B$ that task $A$ and $B$ share the abstract concepts but do not share task-specific knowledge. After fully fine-tuning on task $B$, the model's parameters will somehow over-fit the task-specific knowledge in task $B$ and the abstract concepts stored in these parameters will be partially forgotten.

**Assumption: Identifying the Modularization of Abstract Concepts by Freezing Parameters.** If a module in the model individually store a part of abstract concepts, these parameters can be directly reused in new tasks without further fine-tuning. Furthermore, considering the property above, freezing this abstract module can avoid the forgetting of abstract concepts, resulting in the performance improvement in Main Exp $A \Rightarrow B$.

Based on this assumption, to identify whether abstract concepts are modularized in some parameters, we partially freeze the model in a coarse-to-fine manner. The following experiments are conducted on one of the three pre-training terminal sets. First, we freeze each layer in the model, showing in Figure 8. We find that the last layer in encoder and the first layer in decoder modularize part of abstract concepts. Furthermore, we freeze these two layers in our Contrast Exp $C \Rightarrow B$ and find no improvements in Figure 9a, indicating that the improvement of freezing these two layers comes from keep abstract concepts.

In next step, we try to identify the abstract concepts are stored in attention layers or FF layers. We separately freeze the attention sub-layer and FF sub-layer in the last encoder layer and first decoder layer. Figure 9b shows that attention layers take more responsibility for storing abstract concepts.

Then, we analyze whether these abstract concepts are modularized in some attention heads or averaged in the whole attention layers. Our investigation in Section 6.2 finds that the abstract concepts are centralized in some middle-layer attention heads. Based on the results in Figure 4, we freeze the top

36 heads to further verify that they are responsible for storing abstract concepts. Results in Table 7 indicate that abstract concepts are modularized in this small part of attention heads.

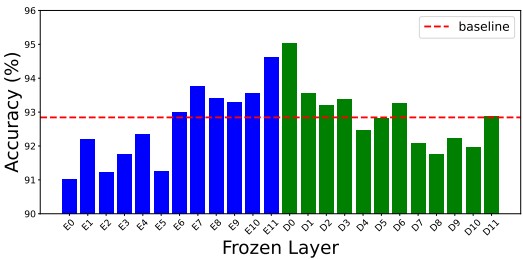

Figure 8: Freeze 24 layers separately in $A \Rightarrow B$.

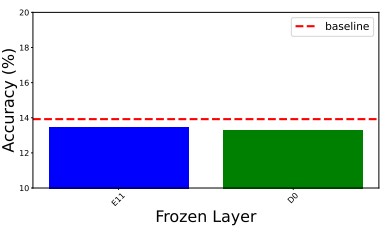

(a) Freeze layers in $C \Rightarrow B$.

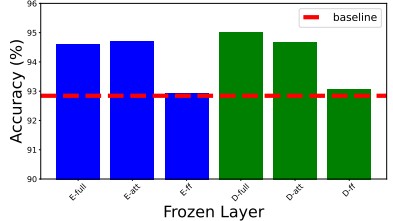

(b) Separately freeze attention and FF in $A \Rightarrow B$.

Figure 9: Further explore the modularity in the middle two layers.

Table 7: Supplement to Table 3. Freeze a set of abstract heads and fine-tune.

|  | Fine-Tuning |
| --- | --- |
| Baseline $A \Rightarrow B$ | 92.8 |
| +Freeze Abstract Heads | 96.6 (+3.8) |
| +Freeze Random Heads | 92.9 |
| -Prune Abstract Concepts | 13.6 |
| Baseline $C \Rightarrow B$ | 13.9 |
| +Freeze Abstract Heads | 14.4 |

## F EXAMPLES IN OUR DESIGNED TASKS

Here, we list some examples in our designed probing tasks. The full sets are contained in our Supplementary Material.

### F.1 GRAMMAR PROBE

Table 8 and 9 shows examples in *Com.* and *Mod.* sub-probes in grammar probe, respectively.

### F.2 OPERATION PROBE

Table 10 lists the operation behind each symbol, and Table 11 shows examples in Conj. sub-probe in operation probe. Figure 10 illustrates the transitions of operations used in contrast task in operation probe.

### F.3 REDUNDANT DESIGNS

For Redundant $\mathbb{G}_s^*$, we supplement the second T-Production rule in Table 1 as "sub / direct-obj / indirect-obj $\twoheadrightarrow \mathbf{S}_{adj} \mathbf{S}_n$". Terminals in $\mathbf{S}_{adj}$ can re regarded as adjectives for $\mathbf{S}_n$. These terminals

Table 8: Examples in different sets in *Com.* sub-probe in grammar probe. The terminal NONE in target side is omitted to more clearly show the structure of the target example.

| Set | Side | Example |
|---|---|---|
| Train Set | Source | Soke incurve huave the soon upon huave a ban bibb huave acetum goladar . |
| | Target | INCURVE ( SOKE ) LG UPON ( SOON ) LG BIBB ( BAN ) LG GOLADAR ( ACETUM ) |
| Transfer Set | Source | Emma liked that a girl saw . |
| | Target | LIKE ( EMMA ) CCOMP SEE ( GIRL ) |
| Test Set | Source | Emma admired that Daniel liked that James meant that a lion froze . |
| | Target | ADMIRE ( EMMA ) CCOMP LIKE ( DANIEL ) CCOMP MEAN ( JAMES ) CCOMP FREEZE ( LION ) |
| Contrast Set | Source | Soke incurve huave the soon upon huave a ban bibb huave acetum goladar . |
| | Target | ( ACETUM ) GOLADAR LG ( BAN ) BIBB LG ( SOON ) UPON LG ( SOKE ) INCURVE |

Table 9: Examples in different sets in mod. sub-probe in grammar probe.

| Set | Side | Example |
|---|---|---|
| Train Set | Source | Safe above the poddy cord the soon a pial . |
| | Target | CORD ( ABOVE ( SAFE , PODDY ) , PIAL , SOON ) |
| Transfer Set | Source | Emma ate the ring beside a bed . |
| | Target | EAT ( EMMA , BESIDE ( RING , BED ) , NONE ) |
| Test Set | Source | The baby on a tray in the house screamed . |
| | Target | SCREAM ( ON ( BABY , IN ( TRAY , HOUSE ) ) , NONE , NONE ) |
| Contrast Set | Source | Safe above the poddy cord the soon a pial . |
| | Target | ( SOON , PIAL , ( PODDY , SAFE ) ABOVE ) CORD |

are only contained in the source side and no terminals in target side are aligned with them. Table 13 shows an example in Redundant $\mathbb{G}_s^*$.

### F.4 DERIVATIONS

Here we show examples of different derivations of original grammars. Table 12 shows examples in different $\mathbb{G}_t^*$. The terminal 'NONE' is omitted.

## G MORE EXPERIMENTAL RESULTS

We present additional experimental results to supplement our probing and analysis.

### G.1 DATA SCALE AND TERMINAL DIVERSITY

Figure 11a shows the effects from data scale for different model sizes. It shows that the performance improves marginally and unstably when the data scale increases from 1.7K to 680K instances. Moreover, it seems that the performance gap between models with different sizes is still considerable when the data scale is enough large.

Figure 11b shows the effects from data diversity for different model sizes. Here, we consider the **terminal diversity** as a perspective of data diversity, i.e., the number of terminals of the grammar. Following Section 4, we only change the number of terminals in $\mathbf{S}_c$ and $\mathbf{S}_C$, increasing from 1 to 128. The overall trend is that the performance improves marginally and unstably when the diversity increases. Interestingly, we observe that for all three models, their performances achieve the peak before rising to 128 terminals and then keep oscillating. We speculate that their performances are bounded by the limited data scale, as we control the data scale as 34K instances when increasing the terminal diversity. To examine our speculation, we conduct another experiment on T5-Base that pre-training on 680K instances with 128 terminals, achieving an average accuracy rate of 93.5%. This performance is higher than the result after pre-training on 680K instances with 32 terminals (89.2%) and higher than the best average accuracy of T5-Base in Figure 11b (88.4%), suggesting that higher data diversity should be equipped with a larger data scale.

Table 10: Symbols and operations in different tasks in operation probe.

| Symbol | Operation in $A$ and $B$ | Operation in $C$ |
|---|---|---|
| a1 | Conjunction | Material Non-implication |
| b2 | Alternative Denial | Material Implication |
| c3 | Disjunction | Converse Non-implication |
| d4 | Joint Denial | Converse Implication |

Table 11: Examples in different sets in operation probe.

| Set | Side | Example |
|---|---|---|
| Train Set | Source | False c3 ( ( ( False a1 ( ( ( False b2 ( True d4 True ) ) d4 True ) d4 True ) ) d4 False ) b2 False ) |
| | Target | True |
| Transfer Set | Source | ( ( False c3 ( False b2 False ) ) b2 True ) d4 ( True b2 ( ( False b2 ( False b2 False ) ) c3 True ) ) |
| | Target | True |
| Test Set | Source | ( True b2 ( ( False a1 False ) d4 False ) ) a1 ( False c3 ( ( ( True a1 True ) c3 True ) a1 True ) ) |
| | Target | False |
| Contrast Set | Source | False c3 ( ( ( False a1 ( ( ( False b2 ( True d4 True ) ) d4 True ) d4 True ) ) d4 False ) b2 False ) |
| | Target | False |

## G.2 MULTI-GRAMMAR PRE-TRAINING

Before this section, we consider the setup in which we only see one pair of input and output grammars during pre-training. This section explores whether multi-grammar pre-training would influence the model to exhibit abstraction. Here, we consider two cases: can or can not access the golden grammar. The golden grammar is the grammar used in the downstream task. For the multi-grammar pre-training, we assemble different target grammars in Section 6.3 while keep the source grammar. During generating pre-training data with more than one target grammar, for each instance, we add a prefix (chosen from *original*, *coarse*, *localreverse*, *nest*, and *reverse*) at the beginning of the source tokens, guiding the model which target grammar it should use. Table 14 shows the results with different ensemble grammars.

First, consider that we have no access to the golden grammar. We take the target grammar that performs the best in Section 6.3, Coarse $\mathbb{G}_t^*$, as the single-grammar baseline. Table 14 shows that augmenting the Coarse $\mathbb{G}_t^*$ with other target grammars can always perform better than the single grammar. Even augmenting with the Reverse $\mathbb{G}_t^*$ from the contrast task can bring a slight gain (1.4% accuracy). It indicates that even though the model has no access to the golden abstract concepts, **increasing the diversity of abstract concepts can make the model better aware of the existence of abstract concepts**. This awareness can be regarded as a higher level of abstraction capibility.

Then, considering that the model has access to the golden grammar, the downstream task performance is lower than only pre-training on the golden grammar (accuracy 88.2%). Therefore, augmenting other similar abstract concepts would confuse the model and make it hard to choose which concepts should be used for the downstream task.

## G.3 INCREASE TOTAL NUMBER OF TERMINALS

We increase the total number of terminals to ~1,500 in our probing tasks and report the results with T5-Base in Table 15. These results are similar to the original results in Table 2 and are still in line with our two expectations.

## H FUZZY GRAMMAR

The abstract concepts discussed in grammar probe in Section 4 can be concretely defined, but in many application scenarios, abstract concepts can be fuzzy (e.g., natural language grammar). Here, we want to examine whether models can learn fuzzy grammar or just can recognize the concrete one.

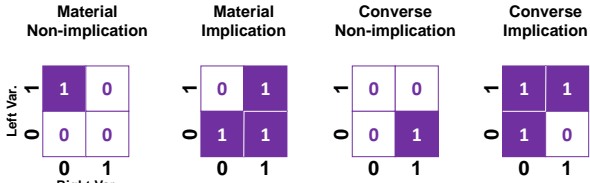

Figure 10: Transitions of operations used in contrast task in operation probe.

Table 12: Examples in different $\mathbb{G}_t^*$.

| Derivation | Target Example |
|---|---|
| Coarse | CLAN ( ) LG INCURVE ( ) LG UPON ( ) |
| LocalR | ( BAN ) CLAN LG ( SOKE ) INCURVE LG ( BIBB, GOALDER, SOON ) UPON |
| Nested | CLAN ( BAN, LG INCURVE ( SOKE, LG UPON ( SOON, GOALDER, BIBB ) ) ) |

We take natural language sentences for experiments and design different sets by mimicking the *Com.* sub-probe in our grammar probe.

Data in our fuzzy grammar probe is taken from Europarl v7 Koehn (2005), a large parallel corpus for machine translation . For the probing on natural language data, we can not guarantee to satisfy the requirements in our framework perfectly, as the grammar of the natural language is hard to be controllable as the formal language. We describe the instantiation of different sets as following.

**Train set $\overline{A}$.** We take the German-to-French (De-Fr) sentence paris as $\overline{A}$.

**Transfer set $\hat{B}$ and Test set $\overline{B}$.** We take English-to-Romanian (En-Ro) as the probing task $B$. As both German and English are belong to the West Germanic language branch while both French and Romanian are belong to Italic branch, the abstract grammars used in De-Fr and En-Ro have some similarities. To satisfy that the $\Uparrow B$ performs poorly, we limit the $\hat{B}$ with only short sentences (15.7/13.8 words in En/Ro sentences in average) while $\overline{B}$ with only long sentences (78.0/74.4 words in En/Ro sentences in average). It means that the model can learn most of the lexicons from $\hat{B}$ but can not be aware of the grammars of long sentences.

**Contrast set $\overline{C}$.** Mimicking the construction of $\overline{C}$ in the formal language task, we also reverse the word order in the target language of $\overline{A}$.

Table 4 shows the performance of two models on natural language data. These results indicate that fuzzy grammar in natural language data can also be learned and transferred by the two PLMs. In addition, as this setting can also be regarded as a length generalization problem, the low $\Delta(C \Rightarrow B)$ further confirm that **our probing results benefit from learning abstract concepts rather than surface patterns (i.e., length distribution)**.

## I   TRY TO MEASURE ABSTRACTION CAPABILITY

As mentioned in Section A that one limitation in our probing is the lack of a metric that can quantitatively measure the abstraction capibility. Thus, we can not compare the abstraction capability of two models with different architectures. Here, we try to design such a metric to compare the abstraction capibility of T5-Base, with ∼220M parameters, and GPT2-Medium, with 345M parameters.

In the beginning, we want to clarify that this metric is just for primary exploration, as it is based on a strong assumption that can not be satisfied in all situations.

**Assumption:** We assume that the performance score of a certain task (such as accuracy and BLEU score) can linearly reflect the ability of the model to solve this task.

It means that, for instance, improving the accuracy from 90% to 100% is not harder than improving from 40% to 50%. Apparently, this assumption does not suit all tasks and performance scores (even

Table 13: Examples in Redundant $\mathbb{G}_s^*$. Gray terminals are redundant that would not be mapped to targets.

| Derivation | Source Example |
|---|---|
| Redundant | A angry pial was ozophen to zogo by odd dermestes . |

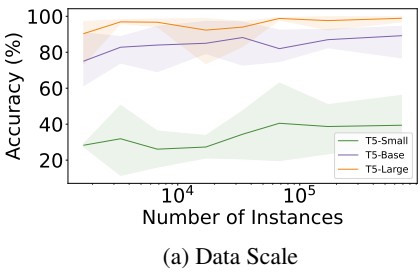

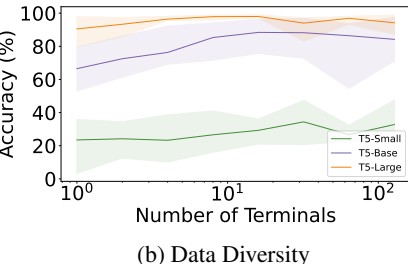

(a) Data Scale

(b) Data Diversity

Figure 11: More results for different data scales and data diversity.

the tasks and scores in our probing). But it does not influence the comparison between T5-Base and GPT2-Medium. The reason will be discussed later.

Intuitively, we consider the contribution of abstract concepts to overall performance as a measure of abstraction capibility, that is,

$$\text{MoA} = \frac{\text{score}_a}{\text{score}_f}, \tag{2}$$

in which the MoA means the **M**etric **o**f **A**bstraction, $\text{score}_f$ means the full performance score on a certain task without limiting the training data, and $\text{score}_a$ means the part of the score contributed by the abstract concepts. Following our probing framework, we consider the $\text{score}_a$ as the relative gain from $\Uparrow B$ to $A \Rightarrow B$. Furthermore, considering the influence of other factors which is reflected by $C \Rightarrow B$, we design the $\text{score}_a$ as:

$$\text{score}_a = \text{score}(A \Rightarrow B) - \max[\text{score}(\Uparrow B), \text{score}(C \Rightarrow B)], \tag{3}$$

in which $\text{score}()$ represents the performance score of a certain procedure, and $\max[\text{score}(\Uparrow B), \text{score}(C \Rightarrow B)]$ means to choose the maximum performance score between $\Uparrow B$ and $C \Rightarrow B$. For the full performance score in the denominator in Equation 2, we evaluate the model performance on $\overline{B}$ after (only) fine-tuning on the full set $\tilde{B}$, which is sampled in the same distribution of $\overline{B}$ (rather than a limited distribution of $\hat{B}$). We denote this procedure as $\Uparrow \tilde{B}$. Thus, the metric in Equation 2 can be formalized as:

$$\text{MoA} = \frac{\text{score}(A \Rightarrow B) - \max[\text{score}(\Uparrow B), \text{score}(C \Rightarrow B)]}{\text{score}(\Uparrow \tilde{B})}. \tag{4}$$

Table 16 shows MoA for two models on grammar probe and fuzzy grammar probe, and lists the scores required to calculate MoA. On each task, MoA of T5-Base is higher than that of GPT2-Medium. Furthermore, during calculating MoA, the baseline score $\max[\text{score}(\Uparrow B), \text{score}(C \Rightarrow B)]$ of T5-Base is always higher than that of GPT2-Medium. As it is harder for the model to improve the accuracy and BLEU scores on these tasks from a relatively higher baseline, MoA can just underestimate the abstract ability of T5-Base. Therefore, we can roughly conclude that the abstraction capibility of T5-Base is higher than GPT2-Medium.

## J    COMPARISON WITH PREVIOUS NEGATIVE RESULTS

Some previous work demonstrated that neural models could not learn abstract concepts (Liu et al., 2020; Chen et al., 2020; Liu et al., 2021a; Chollet, 2019; Mitchell, 2021; Zadrozny, 2021). Our probing results shed some light that neural models, especially PLMs, exhibit abstraction capibility to some extent. Compared with previous work, two points could lead to different conclusions.

Table 14: Downstream performance under different multi-grammar settings.

| | Grammars | | | | Accuracy |
|---|---|---|---|---|---|
| | Reverse | Nest | LocalR | Coarse | |
| | | | | ✓ | 41.6 |
| | ✓ | | | ✓ | 43.0 |
| Without Golden Grammar | | ✓ | | ✓ | 46.4 |
| | | | ✓ | ✓ | 54.6 |
| | ✓ | ✓ | ✓ | ✓ | 55.3 |
| With Golden Grammar | ✓ | ✓ | ✓ | ✓ | 74.0 |

Table 15: Increase total number of terminals.

| Model | Sub-Probe | Control Exp $\Uparrow B$ | Main Exp $A \Rightarrow B$ | Contrast Exp $C \Rightarrow B$ |
|---|---|---|---|---|
| | Avg. | 18.1 | 69.8 (+51.7) | 15.9 (-2.2) |
| T5 | *Com.* | 21.9 | 83.9 | 15.2 |
| | *Mod.* | 14.2 | 55.7 | 16.5 |

The first point is the probing methodology. In all works (including ours), the basic idea of probing abstraction is to separate it with memorization. To implement this idea, previous work has almost always involved designing a special probing task in which memorization of the train set is helpless to solve the test set. However, such an implementation constraints the generation of the train set, which could bring some biases or limitations in training data. To overcome these biases or limitations, the model should have some other abilities more than abstraction, such as reasoning and systematic generalizability. Therefore, the previous disappointing results may have been caused by the lack of other abilities rather than abstraction.

The second point is the test model. Some previous work probed the vanilla Transformer or LSTM while we take the pre-trained language models. We suppose that the model may acquire better abstraction capibility from the pre-training corpus, and can better exhibit this ability with larger model sizes.

# K  DETAILS OF EXPERIMENTS

## K.1  DATA

We show more details about the sets described in Section 4, including data scales, average input lengths and average output lengths.

For the target side grammar of our formal language tasks, we mentioned in Section 4 that we change the original target grammar of COGS to be chain-structured. In Table 18, we list some examples with the original target grammar and the new chain-structured grammar. First, to distinguish the input and output tokens, we capitalize all output tokens (e.g., from "rose" to "ROSE"). Second, we replace the variables (e.g., "x _ 1") in the original grammar with its corresponding terminals (e.g., "ROSE"). Then, we group the terminals of AGENT (e.g., "DOG"), THEME (e.g., "ROSE") and RECIPIENT with their corresponding terminal of PREDICATE (e.g., "HELP") and combine this group of terminals in a function format, i.e., "PREDICATE ( AGENT, THEME, RECIPIENT )". If the predicate is not equipped with an agent, theme or recipient in the original grammar, the corresponding new non-terminals (i.e., AGENT, THEME and RECIPIENT, respectively) in the function format above will be filled with the terminal NONE (e.g., "HELP ( DOG, ROSE, NONE )"). For simplicity, we omitted NONE in Table 1, Table 8, and Table 12. Such a function format is the minimum unit of a CLAUSE. Finally, each CLAUSE is concatenated with another CLAUSE by the terminal CCOMP (e.g., "HOPE ( LIAM, NONE, NONE ) CCOMP PREFER ( DOG, NONE, NONE )").

## K.2  PROCEDURE

**Training**  Each pre-training takes 100,000 steps, and the final-step checkpoint is used for fine-tuning. Each fine-tuning takes 100,000 steps, and the checkpoints for every 10,000 steps are saved.

Table 16: MoA of two models on both grammar probe and fuzzy grammar probe and the scores required to calculate MoA. T5 and GPT2 are T5-Base and GPT2-Medium, respectively.

| Probe | Score | Model | Scores of Procedures | | | | MoA |
|---|---|---|---|---|---|---|---|
| | | | $A \Rightarrow B$ | $\Uparrow B$ | $C \Rightarrow B$ | $\tilde{\Uparrow} B$ | |
| Grammar | Accuracy (%) | T5 | 88.2 | 23.1 | 15.4 | 95.7 | 0.68 |
| | | GPT2 | 48.2 | 1.9 | 2.6 | 93.2 | 0.49 |
| Fuzzy Grammar | BLEU Score | T5 | 35.1 | 24.0 | 26.2 | 41.9 | 0.21 |
| | | GPT2 | 21.0 | 16.4 | 11.6 | 42.2 | 0.11 |

Table 17: Data scales, average input lengths, and average output lengths of different sets in our probing.

| Probe | $\overline{A}$ | | | $\hat{B}$ | | | $\overline{B}$ | | | $\overline{C}$ | | |
|---|---|---|---|---|---|---|---|---|---|---|---|---|
| | Data Scale | Avg Input Len | Avg Output Len | Data Scale | Avg Input Len | Avg Output Len | Data Scale | Avg Input Len | Avg Output Len | Data Scale | Avg Input Len | Avg Output Len |
| Grammar | 34,175 | 16.8 | 29.9 | 24,155 | 9.5 | 10.5 | 1,002 | 34.4 | 76.6 | 34,175 | 16.8 | 29.9 |
| Operation | 100,000 | 16.8 | 1 | 20,000 | 9.5 | 1 | 1,000 | 34.4 | 1 | 100,000 | 16.8 | 1 |
| Fuzzy Grammar | 400,000 | 38.9 | 40.8 | 200,000 | 15.7 | 13.8 | 1,004 | 78.0 | 74.4 | 400,000 | 38.9 | 40.8 |

**Evaluation** We take an early-stopping strategy in our evaluation to avoid catastrophic forgetting. First, each checkpoint saved during fine-tuning is evaluated on the held-out dev set. We choose the first checkpoint that achieves the best dev score for testing. For formal language tasks, we utilize the constraint decoding strategy that the model can only generate the words in the vocabulary.

**Compute and Resources** We majorly use Tesla-V100-16GB GPUs for training and evaluation, except for the experiments on T5-Large or GPT2-Large, which require Tesla-V100-32GB GPUs. On average, one pre-training takes ∼15 GPU hours, one fine-tuning takes ∼15 GPU hours (including saving checkpoints), and one testing takes ∼2 GPU hours (as test cases are very long).

### K.3 HYPERPARAMETERS

Hyperparameters used for training and testing are listed in Table 19.

### K.4 DEFINITION OF PERPLEXITY (PPL)

Th following equation explain how to calculate PPL in Equation 1,

$$\mathrm{PPL}(l_t^i | l_s^i; \theta) = \exp[-\frac{1}{|l_t^i|} \sum_j \log p_\theta(l_t^{i,j} | l_s^i, l_t^{i,<j})]. \tag{5}$$

### K.5 RESULTS

We list the detailed results that are plotted in the figures (i.e., Figure 5 and Figure 6), including the average scores, minimum scores, maximum scores, and standard deviations for all replicate experiments.

Table 18: Examples with the original grammar and the new chain-structured grammar.

| Original Target Grammar | Chain-Structured Target Grammar |
|---|---|
| rose ( x _ 1 ) AND help . theme ( x _ 3 , x _ 1 ) AND help . agent ( x _ 3 , x _ 6 ) AND dog ( x _ 6 ) | HELP ( DOG, ROSE, NONE ) |
| * captain ( x _ 1 ) ; eat . agent ( x _ 2 , x _ 1 ) | EAT ( CAPTION, NONE, NONE ) |
| * dog ( x _ 4 ) ; hope . agent ( x _ 1 , Liam ) AND hope . ccomp ( x _ 1 , x _ 5 ) AND prefer . agent ( x _ 5 , x _ 4 ) | HOPE ( LIAM, NONE, NONE ) CCOMP PREFER ( DOG, NONE, NONE ) |

Table 19: Hyperparameters for training and testing.

| | Grammar Probe | | Operation Probe | |
| | T5 | GPT | T5 | GPT2 |
|---|---|---|---|---|
| Learning Rate | 1e-5 | 1e-5 | 1e-4 | 1e-4 |
| Weight Decay | 0.01 | 0.01 | 0.01 | 0.01 |
| Batch Size | 8 | 8 | 8 | 8 |
| Label Smooth | 0.1 | 0.1 | 0.1 | 0.1 |
| Max Input Len | 1024 | - | 1024 | - |
| Max Output Len | 1024 | - | 1024 | - |
| Max Total Len | - | 1024 | - | 1024 |
| Beam Size | 5 | 5 | 1 | 1 |

Table 20: Detailed results for Figure 5.

| | Input Grammar | | | | | Output Grammar | | | | |
| | Original | Redundant | LocalR | Nest | Reverse | Original | Coarse | LocalR | Nest | Reverse |
|---|---|---|---|---|---|---|---|---|---|---|
| Avg | 88.2 | 86.0 | 76.2 | 55.5 | 15.0 | 88.2 | 41.6 | 33.5 | 29.1 | 15.4 |
| Min | 72.8 | 68.3 | 70.4 | 49.1 | 13.0 | 72.8 | 34.5 | 19.8 | 27.7 | 12.1 |
| Max | 96.9 | 95.6 | 80.6 | 61.1 | 20.1 | 96.9 | 50.8 | 38.7 | 31.7 | 19.2 |
| Std | 8.5 | 10.2 | 4.2 | 4.3 | 2.5 | 8.5 | 4.9 | 6.7 | 1.3 | 2.4 |

Table 21: Detailed results for Figure 6a and 6b.

| | Main Exp | | | | | | Control Exp | | | | | |
| | T5-Small | T5-Base | T5-Large | GPT2 | GPT2-Medium | GPT2-Large | T5-Small | T5-Base | T5-Large | GPT2 | GPT2-Medium | GPT2-Large |
|---|---|---|---|---|---|---|---|---|---|---|---|---|
| Avg | 34.4 | 88.2 | 94.0 | 3.5 | 48.9 | 68.7 | 9.0 | 23.1 | 19.5 | 0.0 | 2.0 | 3.4 |
| Min | 20.7 | 72.8 | 83.2 | 0.0 | 23.9 | 47.2 | 3.0 | 22.6 | 17.8 | 0.1 | 1.1 | 3.1 |
| Max | 47.4 | 96.9 | 97.2 | 12.4 | 63.8 | 85.7 | 6.0 | 23.5 | 21.1 | 0.2 | 2.9 | 3.6 |
| Std | 9.8 | 8.5 | 4.9 | 4.4 | 14.5 | 13.7 | 3.0 | 0.5 | 1.7 | 0.1 | 0.9 | 0.3 |

Table 22: Detailed results for Figure 6c.

(a) T5-Large

| | Data Scale (T5-Large) | | | | | | | |
| | 1.7K | 3.4K | 6.8K | 17K | 34K | 68K | 170K | 680K |
|---|---|---|---|---|---|---|---|---|
| Avg | 90.4 | 96.7 | 96.7 | 92.3 | 94.0 | 98.8 | 97.6 | 98.9 |
| Min | 73.5 | 96.0 | 94.2 | 73.6 | 83.2 | 98.0 | 92.4 | 96.9 |
| Max | 97.1 | 98.1 | 98.5 | 98.7 | 97.2 | 99.3 | 99.8 | 99.7 |
| Std | 10.5 | 0.9 | 1.7 | 10.8 | 4.9 | 0.5 | 3.0 | 1.1 |

(b) T5-Base

| | Data Scale (T5-Base) | | | | | | | |
| | 1.7K | 3.4K | 6.8K | 17K | 34K | 68K | 170K | 680K |
|---|---|---|---|---|---|---|---|---|
| Avg | 75.1 | 82.8 | 84.0 | 85.0 | 88.2 | 82.0 | 87.0 | 89.2 |
| Min | 61.5 | 74.0 | 69.2 | 79.3 | 72.8 | 74.8 | 82.5 | 76.9 |
| Max | 90.8 | 88.7 | 94.4 | 97.2 | 96.9 | 92.3 | 92.7 | 94.4 |
| Std | 10.0 | 4.9 | 8.5 | 6.3 | 8.5 | 6.3 | 3.8 | 7.6 |

(c) T5-Small

| | Data Scale (T5-Small) | | | | | | | |
| | 1.7K | 3.4K | 6.8K | 17K | 34K | 68K | 170K | 680K |
|---|---|---|---|---|---|---|---|---|
| Avg | 28.3 | 31.9 | 26.1 | 27.3 | 34.4 | 40.5 | 38.7 | 39.4 |
| Min | 27.4 | 11.5 | 16.1 | 21.2 | 20.7 | 19.7 | 23.5 | 29.7 |
| Max | 29.2 | 50.5 | 36.2 | 33.7 | 47.4 | 62.9 | 50.9 | 56.0 |
| Std | 0.9 | 15.6 | 9.2 | 4.1 | 9.8 | 19.0 | 9.9 | 9.5 |

Table 23: Detailed results for Figure 11b.

(a) T5-Large

| | Terminal Diversity (T5-Large) | | | | | | | |
|---|---|---|---|---|---|---|---|---|
| | 1 | 2 | 4 | 8 | 16 | 32 | 64 | 128 |
| Avg | 90.5 | 93.3 | 96.4 | 97.9 | 98.1 | 94.0 | 96.9 | 94.2 |
| Min | 79.7 | 85.9 | 95.6 | 95.9 | 97.1 | 83.2 | 93.2 | 87.3 |
| Max | 97.9 | 97.4 | 98.1 | 98.9 | 99.1 | 97.2 | 98.1 | 99.0 |
| Std | 6.9 | 3.9 | 0.9 | 1.0 | 0.7 | 4.9 | 1.7 | 4.6 |

(b) T5-Base

| | Terminal Diversity (T5-Base) | | | | | | | |
|---|---|---|---|---|---|---|---|---|
| | 1 | 2 | 4 | 8 | 16 | 32 | 64 | 128 |
| Avg | 66.5 | 72.5 | 76.3 | 85.3 | 88.4 | 88.2 | 86.4 | 84.2 |
| Min | 53.0 | 61.4 | 69.2 | 71.6 | 75.7 | 72.8 | 54.6 | 71.1 |
| Max | 79.4 | 86.1 | 92.2 | 94.1 | 96.4 | 96.9 | 93.7 | 97.5 |
| Std | 9.8 | 10.1 | 7.8 | 9.4 | 7.6 | 8.5 | 15.2 | 10.7 |

(c) T5-Small

| | Terminal Diversity (T5-Small) | | | | | | | |
|---|---|---|---|---|---|---|---|---|
| | 1 | 2 | 4 | 8 | 16 | 32 | 64 | 128 |
| Avg | 23.5 | 24.2 | 23.3 | 26.6 | 29.3 | 34.4 | 26.7 | 32.8 |
| Min | 3.3 | 12.6 | 10.2 | 16.1 | 21.0 | 20.7 | 22.6 | 24.7 |
| Max | 35.8 | 34.5 | 38.6 | 41.0 | 36.1 | 47.4 | 32.4 | 47.5 |
| Std | 10.1 | 7.5 | 8.3 | 9.2 | 4.9 | 9.8 | 4.3 | 9.6 |

