# OpenReview forum: "Does Deep Learning Learn to Abstract? A Systematic Probing Framework"
_ICLR.cc/2023/Conference — ICLR 2023 poster_

### Official Review · Reviewer_1xjY · 2022-10-21

**Confidence:** 3
**Correctness:** 3
**Technical Novelty And Significance:** 3
**Empirical Novelty And Significance:** 4
**Recommendation:** 10

**Clarity, Quality, Novelty And Reproducibility:**

The paper is moderately clear; I think the presentation could be improved in various ways:
* Showing Table 7 or similar in the main text alongside Table 1 to give a more concrete presentation of qualitatively what individual inputs and outputs like in the different tasks would really help readers to understand the paper.
* I think the clarity would be improved by using the more-standard term “Control experiment” rather than “Controlled experiment”—using “controlled experiment” makes it sound like the main experiments are uncontrolled, and the controlled version is just better.

The quality seems decent.

The work is novel.


**Strength And Weaknesses:**

Strengths:
* The idea is interesting, and the experiments are fairly compelling.
* The thoroughness and design of the experiments is a major plus. Comparing to the contrast and control conditions makes the main experiment results much more compelling.
* I also appreciated the detailed evaluation of different conditions, exploring mutated grammars, analyses of head pruning, scaling, etc. There are a lot of fascinating details in this paper!

Weaknesses:
* It would be great to have a comparison of a model that was not pretrained on natural language first (e.g. T5 initial checkpoint or randomly initialized). That would help clarify the role of that initial pretraining in perhaps teaching the relevant structures of language (and might relate to why these results are different from other transfer results).
* The claim that model scale improves transfer seems very clear. However, the claims that data scale and diversity improve performance seem weakly supported at best. I think the authors should demonstrate that these effects are statistically significant (which appears not at all true from Fig. 6c-d), or remove these claims.
* In fact, the weakness of the data scale effect actually makes me more skeptical of the main results. How can 1.7k pretraining sentences have such a strong effect? How few domain A sentences would be needed to actually see a decrement? How can the model be learning abstract structures from such a small amount of data? It would be useful to clarify this (and/or to see if it breaks down when the scale is reduced farther).

Comments / notes:
* There is some older connectionist work on the idea that neural networks could transfer abstract grammar structures across domains with superficially different features [Dienes et al., 1999]; and some more recent work on generalization of relational properties [Geiger et al., 2022]. They are of course far simpler than the present experiments, but I think these would be a nice piece of motivating context to add from the cognitive side.

References
------

Dienes, Z., Altmann, G. T., & Gao, S. J. (1999). Mapping across domains without feedback: A neural network model of transfer of implicit knowledge. Cognitive Science, 23(1), 53-82.

Geiger, A., Carstensen, A., Frank, M. C., & Potts, C. (2022). Relational reasoning and generalization using nonsymbolic neural networks. Psychological Review. (https://arxiv.org/pdf/2006.07968.pdf)

**Summary Of The Paper:**

This paper studies whether language-pretrained transformer models can abstract grammatical structures and thereby transfer them across superficially unrelated domains. They use carefully controlled experiments, and show some strong transfer effects of grammatical structures, which increase with model scale.

**Summary Of The Review:**

Post-revision update
---------------

The authors have addressed the three main weaknesses I listed in their follow-up experiments, and I believe they have improved the clarity of the paper. I have improved my score accordingly.

Original review
--------------------

I think this paper is a potentially-compelling demonstration of a fascinating transfer phenomenon. I think this is a very interesting work that should lead to lots of follow-on investigations. I have some lingering concerns and suggestions noted above, that I hope the authors will address, and that I expect would increase the impact of the paper.

---

> ### Author Response · Authors · 2022-11-16
> **Response to Reviewer 1xjY**
>
> Thanks for your comments and concrete suggestions.
>
> **Comparison with randomly initialized models.**
>
> We examine the randomly initialized T5 and GPT2 models (Section 6.4 and Table 5 of the revised paper).
> As expected, they do not exhibit abstraction capability in our probe.
> This indicates that initial pre-training takes a critical role for abstraction.
> Take T5-Base as an example:
>
> | Performance Gain                 | Rand-Init | Pre-Trained |
> |----------------------------------|-----------|-------------|
> | $\small \Delta(A\Rightarrow{}B)$ | **+2.0**  | **+53.2**   |
> | $\small \Delta(C\Rightarrow{}B)$ | +1.1      | -2.7        |
>
> Recap that task *A* and *B* share the same abstract concepts, and the Expectation 1 in our framework is that $\small \Delta(A\Rightarrow{}B)$ should be significantly high.
> Although the rand-init model can still achieve >98% dev set accuracy on task *A*,
> the low transfer gain from *A* to probing task *B* suggests that it just learned surface patterns rather than abstract concepts in task *A*.
> The different behaviors of rand-init models and pre-trained models on $\small\Delta (A\Rightarrow{}B)$ indicate that initial pre-training takes a critical role for 'teaching' the rand-init models to mine the inner structure of the language.
> See L333-339 in the revision for details.
>
> **Make our claims in Section 6.4 more statistically significant.**
>
> - For data scale, we add two points with lower data scales (~1k and ~300) to Figure 6c, thus making our claim more significant.
> We also revise L328-332 to describe the performance trend.
>
> - For data diversity, we conduct more experiments and move it to Appendix G.1 (due to the page limitation).
> Our claim of data diversity is more significant under larger data scale.
> For example, when data scale is ~680k, diversity(128) achieves an accuracy of 93.5%, which outperforms that of diversity(32) (89.2%).
>
> **How few domain *A* sentences would be needed to actually see a decrement? How can a few thousand examples have such a strong effect?**
>
> - The performance drops dramatically when the data scale is below 1.7k-3.4k, as shown in Figure 6c in our revision.
>
> - A few thousand examples are sufficient to cover the ~200 possible structures in our FLT task.
> That is why such few thousand examples could have strong effect.
>
> **Comments about writing:**
>
> - (1) We add your recommended motivating context from the cognitive side in L87-90.
> - (2) We take the term "Control experiment" as you suggested.
> - (3) Due to page limitation, we included some concrete examples of our tasks in Appendix F.

---

> > ### Comment · Reviewer_1xjY · 2022-11-16
> > **Thanks for the updates, I think these improve the paper**
> >
> > I appreciate the additional experiments! I think this paper is a good contribution, and I will revise my review to note the improvement.

---

### Official Review · Reviewer_tkML · 2022-10-23

**Confidence:** 3
**Correctness:** 3
**Technical Novelty And Significance:** 1
**Empirical Novelty And Significance:** 2
**Recommendation:** 6

**Clarity, Quality, Novelty And Reproducibility:**

The paper is presents its framework and experiments clearly; results are, in the opinion of the reviewer, mostly concerned with transferability than abstraction. Setup is explained and data shared.

**Strength And Weaknesses:**

The paper considers a very interesting and relevant problem, and it explores it using a sensible framework and experiments. However, the reviewer is uncertain about its conclusions conceptually and pragmatically. From a conceptual point of view, abstraction is never well defined; all the empirical simulations seem to me deal with transfer learning, thus leading to a sort of identification of abstraction = transferability. I assume (together with some of the authors referenced in the introduction) that abstraction is something more than transfer ability. The author should probably discuss this point. From a pragmatic point of view, even if the authors explored some different settings in Section 6.4, it seems that the cases considered are limited, and hence the generalizability of the results.
Furthermore, I think it would be interesting if the authors were to engage with other relevant papers. For instance, the framework considered by the authors is built similarly to other common datasets (e.g.: ARC by Chollet); beyond modality, how are the two related? The "memorize-and-abstract" is also reminiscent of the learning dynamics in information bottleneck (Tishby); is there an analogy there?

**Summary Of The Paper:**

The paper introduces a framework to evaluate the abstraction capability of deep models in terms of transferability. A number of empirical simulations are run on language models in order to evaluate the dynamics of abstraction.

**Summary Of The Review:**

The paper offers a useful probing framework for assessing the learning capabilities of deep neural models. Results are interesting in terms of transfer learning, but their meaning in terms of abstraction may be debatable. Explaining this gap and relating to other literature on abstraction would strongly improve the contribution.

---

> ### Author Response · Authors · 2022-11-16
> **Response to Reviewer tkML**
>
> Thanks for your comments and concrete suggestions.
>
> **Conceptual concern: the relation between abstraction and transferability.**
>
> Briefly, in our work, the goal is to systematically explore 'abstraction', while 'transferability' is one tool to facilitate our exploration rather than being a research target.
> In ML, when we want to verify whether a model has learned some skills, we always apply the model to tackle new test tasks from either the same domain (namely in-domain learning) or another domain (i.e., transfer learning).
> In this paper, we use 'transferability' as a tool to set up a controlled and constrained test suit to reveal whether a model has learned some abstract concepts.
>
> Concretely, to examine whether the abstract concepts can be learned, we check whether they can be reused (transferred) in a constrained setting where the transfer of surface patterns are explicitly avoided.
> After observing a good transferability, we further confirm that this transferability is from abstraction rather than other influential factors by taking Control Exp and Contrast Exp.
> Finally, the successful transfer shown by $\small \Delta(A\Rightarrow{B})$ and the failed transfer shown by $\small \Delta(C\Rightarrow{B})$ jointly demonstrate the abstraction capability.
>
> In other words, rather than studying the effectiveness of the transfer learning, our main goal is to give an explicit demonstration of whether the abstract concepts can be learned.
>
> **Pragmatic concern about generalizability.**
> - Besides the observations in Section 5 and 6, one of our key contributions is the probing framework (Section 3) that can generalize to more models and tasks, thus facilitating researchers in different fields to study and understand abstraction capabilities systematically.
>
> - Besides the FLT tasks and the analysis in Section 6.4, we also have another set of probing tasks in Appendix D, following our probing framework.
> We hope it can further demonstrate the generalizability of our framework.
>
> **Relation with ARC dataset.**
>
> ARC and our work consider some similar high-level principles, such as alleviating influence from surface patterns (called 'task-specific skill' in ARC).
>
> Beyond modality, the main difference is that ARC considers the in-task probe while we leverage the cross-task probe.
>
> - The in-task probe cannot explicitly decouple the effects from surface patterns and abstract concepts, thus ARC considers an implicit approach that injects great task diversity to increase the difficulty of learning 'practical shortcuts'.
> - Our framework leverages the cross-task setting along with controlled and constrained designs to directly decouple the effects from surface patterns and abstract concepts.
> Moreover, we explicitly check for potential performance leakage from surface patterns by comparing performance in different (while well-controlled) experimental settings.
>
> We add these comparisons in L107-118.
>
> **Relation with information bottleneck.**
>
> We add the following paragraph to Appendix B:
>
> [1] found a two-phase learning process from the view of information flow in deep neural networks:
> empirical error minimization phase and representation compression phase.
> This process is different from the memorize-then-abstract process since they measure the training dynamics in quite different perspectives.
> The former focuses on the compression of representation (and reduction of mutual information) while the latter portrays the learning of abstract concepts.
> The analogy between the two may lie in that the extraction of abstract concepts from concrete instances is in some way have the effect of information compression.
>
> [1] Shwartz-Ziv, Ravid, and Naftali Tishby. "Opening the Black Box of Deep Neural Networks via Information."

---

> > ### Comment · Reviewer_tkML · 2022-11-21
> > **Response to the authors**
> >
> > I appreciate the review of the authors addressing in particular the relation with ARC and bottleneck theory.
> >
> > I am still a little bit uncertain with the dealing of 'abstraction', in the sense that its meaning remains at most vague, and a project aimed at a 'systematic exploration of abstraction' would require a precise notion of it. I agree, though, that transferability may be seen as a (necessary?) condition for abstraction, and I definitely recognize the value of the work done. I think this research could improve even further in the future by relying on more rigorous definitions or framework for abstraction (e.g. something similar to what is done in [https://arxiv.org/abs/2106.02997]). I reviewed my recommendation.

---

### Official Review · Reviewer_dxgs · 2022-10-24

**Confidence:** 3
**Correctness:** 4
**Technical Novelty And Significance:** 4
**Empirical Novelty And Significance:** 4
**Recommendation:** 8

**Clarity, Quality, Novelty And Reproducibility:**

The paper is clearly written, and the results are novel (to the best of my knowledge).

**Strength And Weaknesses:**

## Strengths:
- The design of the probing framework includes a number of careful control experiments.
- The analyses cover a wide range of factors that might contribute to the emergence of abstract concepts.

## Weaknesses:
- A key claim is that *pre-trained* language models, according to these results, show evidence of being able to learn abstract concepts. But the paper does not directly evaluate the importance of this pre-training (i.e. the generic pre-training that occurs prior to the experiment-specific pre-training in their probing framework). This should be straightforward to test, by simply using the same architectures (T5 and GPT-2), but training them from scratch on the probing framework training sets. I strongly suspect that the broader background pre-training is very important for this capacity, but it would be good to demonstrate this.

### Other comments:
- In the related work, the authors make a distinction between previous 'in-task' evaluations of abstraction vs. the proposed (between-task) evaluation. Are these really fundamentally different? Is there even a principled way to demarcate what constitutes a 'task'? It would be good for the authors to discuss this issue a bit more.
- The 'memorize-then-abstract' phenomenon is interesting, but since the 'memorization' component of this phenomenon involves performance on a heldout (in-task) test set, it seems like 'interpolate-then-abstract' may be a better term, since it isn't strictly speaking memorization.
- In the analysis of modularization, the paper states that 'we consider the first 36 heads in Figure 4', which are also described as 'abstract heads'. Does 'first' mean the heads with the highest DPC? The way it is phrased makes it sound as though the heads with the lowest value along the X axis were chosen, which would seem to be heads that are not important for abstraction.
- In Figures 6a and 6b, it would be helpful to include the arrow notation (i.e. $A \Rightarrow B$) in the legend.
- The memorize-then-abstract phenomenon seems potentially related to 'grokking' [1], in that both involve improvement in generalization after a model has already largely converged on the training data.

[1] Power, A., Burda, Y., Edwards, H., Babuschkin, I., & Misra, V. (2022). Grokking: Generalization beyond overfitting on small algorithmic datasets. arXiv preprint arXiv:2201.02177.

**Summary Of The Paper:**

This paper proposes a novel benchmark for evaluating whether neural networks have learned abstract concepts. This benchmark is applied to pre-trained language models, revealing evidence that they indeed can learn abstract concepts. The paper also presents a range of analyses assessing the factors that contribute to the emergence of abstract concepts.

**Summary Of The Review:**

This paper provides an interesting analysis of the emergence of abstract concepts in pre-trained language models, with a set of well-controlled experiments and extensive analyses. The only major missing element is an analysis of the extent to which pre-training is necessary for the emergence of abstract concepts.

Update after rebuttal: the authors have added new experiments demonstrating the importance of pre-training. This addresses my only major concern with the work, and I have updated my score accordingly.

---

> ### Author Response · Authors · 2022-11-16
> **Response to Reviewer dxgs**
>
> Thanks for your comments and detailed suggestions.
>
> **Directly evaluate the importance of generic pre-training.**
>
> We examine the T5 and GPT2 models without loading pre-trained checkpoints (Section 6.4 and Table 5 of the revised paper).
> As expected, they do not exhibit abstraction capability in our probe.
> This indicates that initial pre-training takes a critical role for abstraction.
> Take T5-Base as an example:
>
> | Performance Gain                 | Rand-Init | Pre-Trained |
> |----------------------------------|-----------|-------------|
> | $\small \Delta(A\Rightarrow{}B)$ | **+2.0**  | **+53.2**   |
> | $\small \Delta(C\Rightarrow{}B)$ | +1.1      | -2.7        |
>
> Recap that task *A* and *B* share the same abstract concepts, and the Expectation 1 in our framework is that $\small \Delta(A\Rightarrow{}B)$ should be significantly high.
> Without pre-trained checkpoints, although the model can still achieve >98% dev set accuracy on task *A*,
> the low transfer gain on probing task *B* suggests that it just learned surface patterns rather than abstract concepts from task *A*.
> These results demonstrate that broader background pre-training is critical for the emergence of abstraction capability.
> See L333-339 in the revision for details.
>
> **Other comments:**
>
> - (1) The fundamental difference between the 'in-task' probe and the 'cross-task' probe is that the former cannot directly and explicitly avoiding the potential influence of task-specific characteristics, while the latter can naturally avoid this by designing cross-task settings with different (while controlled) characteristics.
> We make this clearer in L107-108.
> - (2) The term 'memorize' in this work refers to 'memorizing surface patterns' rather than 'memorizing training examples'.
> We make this clearer by highlighting this definition in L225 and L232.
> - (3) We revise our wording in L266-267 to clarify that we choose the attention heads with the highest DPC.
> - (4) We add notions to the legend in Figure 6a and 6b.
> - (5) Both the 'grokking' and 'memorize-then-abstract' phenomena indicate that some general concepts can be learned in a later training stage.
> The difference is that the 'grokking' focuses on generalization beyond over-fitting training data, while 'memorize-then-abstract' portrays the transfer of abstract concepts beyond task-specific surface patterns.
> We include these comparisons in Appendix B.

---

> > ### Comment · Reviewer_dxgs · 2022-11-19
> > **Reply**
> >
> > Thanks very much to the authors for sharing these additional results, and also for the clarifications to the manuscript. My concerns are satisfactorily addressed, I will update my score accordingly.

---

### Official Review · Reviewer_VyFa · 2022-10-29

**Confidence:** 3
**Correctness:** 3
**Technical Novelty And Significance:** 3
**Empirical Novelty And Significance:** 3
**Recommendation:** 8

**Clarity, Quality, Novelty And Reproducibility:**

The paper is well-organized, and the study is extensive. The experimental designs are new and results are insightful.

**Strength And Weaknesses:**

Strength
======
- An extensive and well-designed study.
- Results are revealing. Limitations are articulated and partly addressed.
- In-depth analysis is provided. The difference from previous negative results on the generalization of neural networks is briefly discussed.

Weaknesses
==========
- The number of terminals is a bit small compared to real languages.

**Summary Of The Paper:**

The paper empirically investigates whether pretrained large language models can have abstraction capability --  whether the model can be aware of the grammar instead of memorizing surface word patterns, taking from a transferability perspective between tasks of different characteristics. The paper also reports the effect of the training dynamics on the abstraction. Under this scheme, interesting results were found:  (1) there is a "memorize-then-abstract" two-stage process in the training time; (2) several middle-layer heads capture the abstract concepts, and (3) scaling does help.


**Summary Of The Review:**

A nice addition to the recent literature of large-language models, helping to explain their recent great performance on different settings.

---

> ### Author Response · Authors · 2022-11-16
> **Response to Reviewer VyFa**
>
> Thanks for your comments and concrete suggestions.
>
> **Increase the total number of terminals in our probe.**
>
> We add a set of experiments increasing the total number of terminals to ~1,500 (Appendix G.3).
> The results are similar to the original ones, indicating that our results can generalize to larger vocabularies.
> We briefly show the results as follows:
>
> | Performance Gain                 | More Terminals | Original Results |
> |----------------------------------|----------------|------------------|
> | $\small \Delta(A\Rightarrow{}B)$ | +51.7          | +53.2            |
> | $\small \Delta(C\Rightarrow{}B)$ | -2.2           | -2.7             |
>
> Results with more terminals are still in line with the two expectations on performance gains in our framework.
>
> We hope that these results will strengthen our probe and address your concerns.

---

### Decision · Program_Chairs · 2023-01-20

**Decision:**

Accept: poster

**Justification For Why Not Higher Score:**

My concern with the paper is largely that the English is poor and it's not very clearly written. I could be convinced to push it up to spotlight or oral.

**Justification For Why Not Lower Score:**

The next lower score is "reject" which is certainly not the right outcome for this paper.

**Metareview: Summary, Strengths And Weaknesses:**

The authors present a framework for systematically evaluating whether a language model can learn abstract concepts (syntactic categories) when fine-tuned on a target task. In this case the target task is one whose language has a grammar homomorphic to the source grammar, which ensures that the two tasks share the same abstract concepts. This framework is used to probe whether, after fine-tuning, the model is able to solve a task which a model trained on the fine-tuning dataset alone cannot, demonstrating by transfer the existence of some ability to employ abstractions.

Strengths: All the reviewers agreed that the paper provides a nice, well-controlled set of experiments to investigate the learning of these kinds of abstract categories.

Weaknesses:

1. I share the concern with reviewer tkML that the term "abstraction" is not very clearly articulated (e.g. line 52). My interpretation was specifically of an abstract category (a set of phrases in a language corresponding to a grammar non-terminal). Perhaps the authors can clarify this in the final version. The "formal" definition given on p. 152 is not formal at all (just sort of formally stated).  My initial concern while reading through was with that the results were not sufficiently general, but after discussion with the reviewers and given the the experiments on grammar mutations, which show a robust degradation in performance when the source and target grammars are not perfectly homomorphic, my worry was largely allayed.

2. The writing is not very clear and could use editing.  A few minor issues:
-line 162: grammar is a set of syntactic rules; in your setting those rules are hidden
table 1: you say you use a double-headed arrow instead of a single-headed one (usual grammar rule) but don’t say how your notation differs.
-why not just say that your source and target tasks require languages with homomorphic grammars?
-line 255: say what “DPC” stands for
-line 266: A module (not "modular") is a subset of the parameters …
The conclusion is a bit too short to be helpful (e.g. “insightful conclusions”). Perhaps just drop it.

The paper specifically focuses on a fine-tuning setting for a pre-trained model. This shows that abstractions can be induced by fine-tuning when the tasks are sufficiently similar. The reviewers asked for additional experiments to evaluate the importance of pre-training which the authors performed. The results show pre-training is essential but don't say much about the nature of this kind of latent ability to be fine-tuned to learn abstraction or what mechanisms are at play in eliciting the abstractions through fine-tuning. This will be interesting to explore in future work.


**Note From Pc:**

if the above contains the word "oral" or "spotlight" please see: "oral" presentation means -> notable-top-5% and "spotlight" means -> notable-top-25%. As stated in our emails, we are disassociating presentation type from AC recommendations

**Summary Of Ac-Reviewer Meeting:**

N/A